# MicroRNA-483 amelioration of experimental pulmonary hypertension

Jin Zhang[1,†], Yangyang He[2,†], Xiaosong Yan[1], Shanshan Chen[1], Ming He[3], Yuyang Lei[1], Jiao Zhang[1,3,4], Brendan Gongol[3], Mingxia Gu[5], Yifei Miao[5], Liang Bai[1], Xiaopei Cui[6], Xiaojian Wang[2], Yixin Zhang[2], Fenling Fan[4], Zhao Li[1], Yuan Shen[7], Chih-Hung Chou[8] [iD], Hsien-Da Huang[9], Atul Malhotra[3], Marlene Rabinovitch[5], Zhi-Cheng Jing[10,*] [iD] & John Y-J Shyy[1,3,**] [iD]

## Abstract

Endothelial dysfunction is critically involved in the pathogenesis of pulmonary arterial hypertension (PAH) and that exogenously administered microRNA may be of therapeutic benefit. Lower levels of miR-483 were found in serum from patients with idiopathic pulmonary arterial hypertension (IPAH), particularly those with more severe disease. RNA-seq and bioinformatics analyses showed that miR-483 targets several PAH-related genes, including transforming growth factor-β (TGF-β), TGF-β receptor 2 (TGFBR2), β-catenin, connective tissue growth factor (CTGF), interleukin-1β (IL-1β), and endothelin-1 (ET-1). Overexpression of miR-483 in ECs inhibited inflammatory and fibrogenic responses, revealed by the decreased expression of TGF-β, TGFBR2, β-catenin, CTGF, IL-1β, and ET-1. In contrast, inhibition of miR-483 increased these genes in ECs. Rats with EC-specific miR-483 overexpression exhibited ameliorated pulmonary hypertension (PH) and reduced right ventricular hypertrophy on challenge with monocrotaline (MCT) or Sugen + hypoxia. A reversal effect was observed in rats that received MCT with inhaled lentivirus overexpressing miR-483. These results indicate that PAH is associated with a reduced level of miR-483 and that miR-483 might reduce experimental PH by inhibition of multiple adverse responses.

**Keywords** miR-483; endothelium; pulmonary hypertension; TGF-β
**Subject Categories** Molecular Biology of Disease; Respiratory System; RNA Biology

## Introduction

Pulmonary arterial hypertension (PAH) is a multifaceted vascular disease resulting from functional changes and structural remodeling in small pulmonary arteries (Hoeper *et al*, 2013). In patients with PAH and various animal models with experimental pulmonary hypertension (PH), endothelial dysfunction is linked to increased production of vasoconstrictors, pro-inflammatory cytokines, and growth factors together with attenuated production of vasodilators (Schermuly *et al*, 2011). The current drug therapy for PAH, including bosentan, sildenafil, macitentan, riociguat, prostacyclin, and its derivatives, aims to target pathways related to endothelin-1 (ET-1), nitric oxide, and vasoconstrictor prostaglandins (Galie *et al*, 2016a,b). However, these agents improve quality of life and longevity but are not curative. MicroRNAs (miRNAs) modulate gene expression by inhibiting translation or inducing degradation of target mRNA transcripts. New therapeutic approaches involving miRNA therapeutics are emerging that either mimic or inhibit miRNAs (Janssen *et al*, 2013).

MicroRNA-483 (miR-483), containing both miR-483-3p and miR-483-5p (hereafter referred to as miR-483-3p/-5p), are intronic miRNAs that are encoded together with insulin-like growth factor 2

1  Cardiovascular Research Center, School of Basic Medical Sciences, Xi'an Jiaotong University Health Science Center, Key Laboratory of Environment and Genes Related to Diseases, Ministry of Education of China, Xi'an Jiaotong University, Xian, China
2  State Key Laboratory of Cardiovascular disease & FuWai Hospital, Chinese Academy of Medical Sciences & Peking Union Medical College, Beijing, China
3  Department of Medicine, University of California, San Diego, La Jolla, CA, USA
4  Department of Cardiology, First Affiliated Hospital, Xi'an Jiaotong University, Xian, China
5  Department of Pediatrics (Cardiology), Cardiovascular Institute and Wall Center for Pulmonary Vascular Diseases, Stanford University School of Medicine, Stanford, CA, USA
6  Department of Geriatric Medicine, Qilu Hospital of Shandong University, Jinan, China
7  Department of Epidemiology and Health Statistics, School of Public Health, Xi'an Jiaotong University, Xian, China
8  Department of Biological Science and Technology, National Chiao Tung University, Hsinchu, Taiwan
9  Warshel Institute for Computational Biology, School of Life and Health Sciences, The Chinese University of Hong Kong, Shenzhen, China
10 Department of Cardiology & Key Lab of Pulmonary Vascular Medicine, Peking Union Medical College Hospital, Chinese Academy of Medical Sciences and Peking Union Medical College, Beijing, China
    *Corresponding author. Tel: +86 1069 155023; E-mail: jingzhicheng@vip.163.com
    **Corresponding author. Tel: +1 858 534 3736; E-mail: jshyy@ucsd.edu
    †These authors contributed equally to this work as first authors

**Table 1. Clinical characteristics of PAH patients and control subjects.**

| | Serum | | CD144-EVs | |
|---|---|---|---|---|
| | Control ($n$ = 95) | IPAH ($n$ = 139) | Control ($n$ = 34) | IPAH ($n$ = 37) |
| Sex [female, $n$ (%)] | 77 (81%) | 117 (84%) | 3 (82%) | 5 (86%) |
| Age (years) | 33 (21–47) | 32 (17–70) | 37 (20–46) | 35 (23–40) |
| Functional class [$n$ (%)] | | | | |
| I | – | 3 (2%) | – | 2 (5%) |
| II | – | 21 (42%) | – | 16 (43%) |
| III | – | 23 (49%) | – | 16 (43%) |
| IV | – | 4 (7%) | – | 3 (8%) |
| CI (l/min/m$^2$) | – | 2.3 ± 0.6 | – | 2.2 ± 0.5 |
| PVR (Wood U) | – | 14.1 ± 6.2 | – | 14.2 ± 6.2 |
| mPAP (mmHg) | – | 60.0 ± 17.2 | – | 58.7 ± 15.5 |
| RAP (mmHg) | – | 8.1 ± 4.8 | – | 7.9 ± 5.0 |
| SvO2 (%) | – | 66.5 ± 8.7 | – | 66.2 ± 8.1 |
| NT-proBNP (ng/l) | – | 1683 ± 2348 | – | 1933 ± 2393 |
| 6MWD (m) | – | 444.5 ± 102.7 | – | 436.9 ± 102.7 |
| Medication [$n$ (%)] | | | | |
| ERA | – | 67 (48.2%) | – | 22 (59.5%) |
| PDE5 inhibitor | – | 79 (56.8%) | – | 27 (73.0%) |
| Epoprostenol | – | 26 (18.7%) | – | 7 (18.9%) |

(*IGF2*) (Li *et al*, 2015). The sequences of miR-483-3p/-5p are highly conserved among mammalian species including human, mouse, and rat. MiR-483 targets the 3′UTR of connective tissue growth factor (CTGF), platelet-derived growth factor-β (PDGF-β), tissue inhibitor of metalloproteinase 2, and Smad4, a key component protein in the transforming growth factor β (TGF-β) signaling pathway (Li *et al*, 2014; Anderson & McAlinden, 2017; He *et al*, 2017). Additionally, miR-483 suppresses the growth and proliferation of several cell types by downregulating extracellular signal-regulated kinase 1, mitogen-activated protein kinase-activated protein kinase 2, Yes-associated protein 1, and IGF1 (Bertero *et al*, 2011; Wang *et al*, 2012; Ni *et al*, 2013). With respect to endothelial biology, we previously showed that miR-483 enhances endothelial cell (EC) function, in part via its anti-fibrogenic effect and a recent report by others shows that disturbed flow decreases miR-483 level in ECs (He *et al*, 2017; Fernandez Esmerats *et al*, 2019).

Based on this conceptual framework, we sought to test the hypothesis that miR-483 is pathophysiologically important in PH and that it may be useful as a disease biomarker and a therapeutic target. In the current study, we investigated whether low levels of miR-483 were associated with idiopathic PAH (IPAH). Our results show that the levels of circulating miR-483 were indeed lower in IPAH patients and seemed to correlate inversely with the severity of clinical manifestations. Experimental validation revealed that miR-483 targeted multiple genes including TGF-β, interleukin 1β (IL-1β), and ET-1 in ECs. Rats with miR-483 overexpression in the endothelium were PH-protected. Of translational relevance, intratracheal administered miR-483 reversed monocrotaline (MCT)-induced PH in rats, which provides therapeutic potential of miR-483.

# Results

### Decreased miR-483 level in human IPAH patients

We first examined the circulatory levels of miR-483-3p/-5p in a cohort of IPAH patients (patient demographics and clinical characteristics in Table 1). Compared to age- and sex-matched healthy controls (HCs), the circulating levels of miR-483-3p/-5p were significantly lower in IPAH patients (Fig 1A). The receiver operating characteristic (ROC) curves shown in Fig 1B reveal that patients with miR-483-3p and miR-483-5p levels < 27% and 26% of those in HCs might have PAH, with sensitivity 88.4% and 82.1%; specificity 56.8% and 48.9% (area under the ROC curve, 0.77 and 0.66, respectively). According to the new PAH risk stratification for prognosis and initial therapy (2018 World Symposium on Pulmonary Hypertension; Galie *et al*, 2019), we divided IPAH patients into low-, intermediate-, and high-risk groups with estimated 1-year mortality risks of < 5%, 5–10%, and > 10%, respectively. Of note, miR-483-3p level was lower in intermediate- and high-risk than low-risk patients (Fig 1C).

Patients with various forms of PAH, including IPAH, have higher serum or plasma levels of ET-1 and IL-6 (Schermuly *et al*, 2011). In the present study, the serum levels of ET-1 and IL-6 in the IPAH cohort were also significantly higher than those in control groups (Appendix Fig S1A and B) with areas under the ROC curve 0.74 and 0.74, respectively (Appendix Fig S1C and D). Interestingly, the serum levels of ET-1 and IL-6 seemed to be inversely correlated with those of miR-483-3p (Fig 1D).

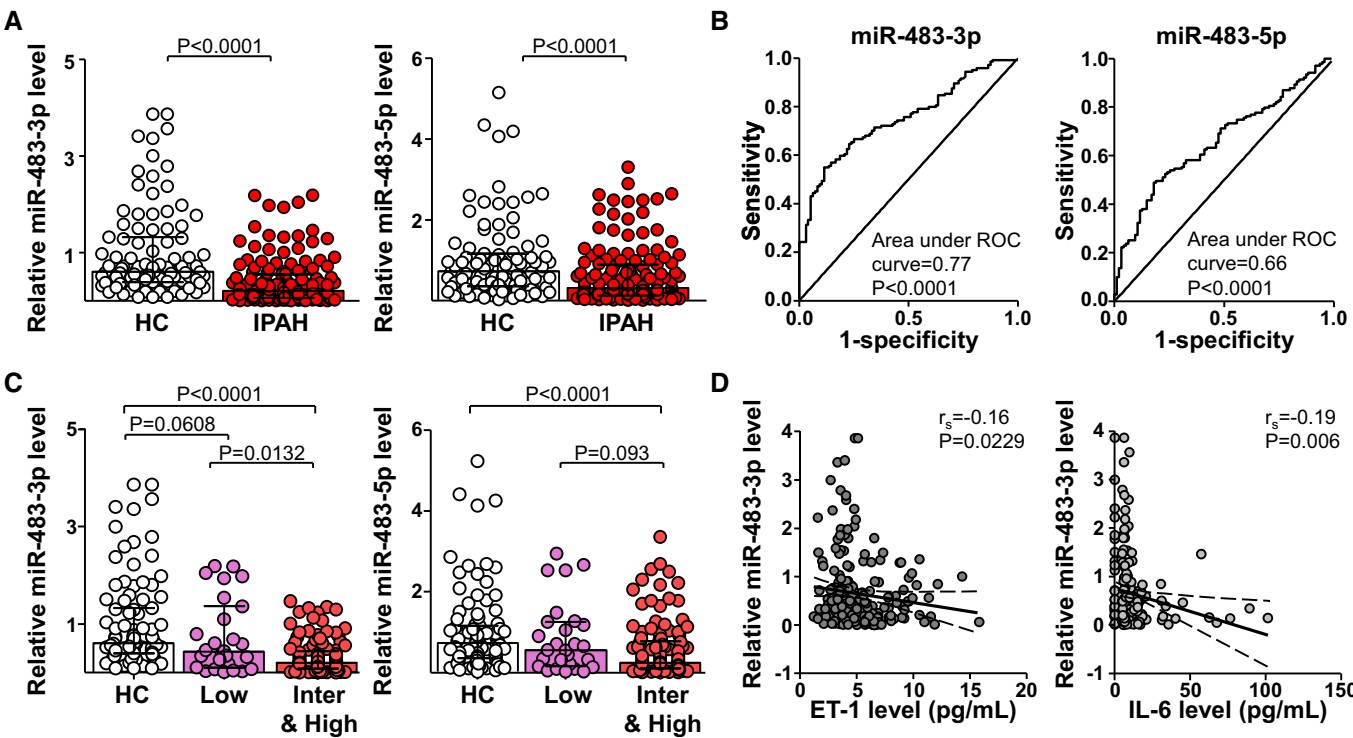

**Figure 1. Lower serum miR-483 level in IPAH patients.**

A   Serum levels of miR-483-3p/-5p in IPAH patients (*n* = 139) and HC (*n* = 95) measured by qPCR. The data are fold change normalized to the averaged level of HC.

B   ROC curve with sensitivity and specificity of serum levels of miR-483-3p/-5p for differentiating IPAH patients from HCs at diagnosis (IPAH, *n* = 139; HC, *n* = 95).

C   Levels of miR-483-3p/-5p associated with PAH risk in three groups. IPAH patients were divided into a low-risk group (Low) and an intermediate- plus high-risk group (Inter&high) according to the World Symposium on Pulmonary Hypertension 2018 [14].

D   Levels of miR-483-3p were inversely correlated with serum levels of ET-1 (IPAH, *n* = 118; HC, *n* = 95) and IL-6 (IPAH, *n* = 112; HC, *n* = 93).

Data information: Values are expressed as median ± interquartile range. Statistical test: Mann–Whitney *U*-test (A), Kruskal–Wallis *U*-test (C).
Source data are available online for this figure.

## miR-483 targets PAH-related genes and pathways in endothelium

We postulated that the lower levels of miR-483-3p/-5p in IPAH patient sera were linked to their decreased expression in pulmonary endothelium. To confirm this notion, we isolated CD144 (VE-cadherin, an EC marker)-enriched extracellular vesicles (EVs) from serum of IPAH patients and HCs. As illustrated in Fig 2A, miR-483-3p/-5p content was lower in CD144-enriched EVs from IPAH than HC. The lower levels of miR-483 found in circulation and CD144-enriched EVs of IPAH patients suggested that the expression of miR-483-3p/-5p might affect genes and pathways involved in PAH.

We conducted RNA-seq analysis to examine the transcriptomes in cultured human pulmonary arterial ECs (PAECs) overexpressing miR-483. As illustrated in Fig 2B and Appendix Fig S2, gene ontology (GO) analysis of differentially regulated pathways showed that miR-483 was correlated with negative regulation of Wnt signaling and inflammatory response, TGF-β receptor signaling, cell adhesion, response to hypoxia, apoptotic processes, oxidation–reduction processes, and negative regulation of cell migration and proliferation. Such changes in transcriptomes suggested that miR-483-3p/-5p could affect the EC phenotype. The heatmap shown in Fig 2C revealed that miR-483 overexpression downregulated genes involved in cell proliferation, migration, inflammation, Wnt and

TGF-β receptor signaling, apoptosis process, response to hypoxia, and oxidative stress. In contrast, genes (e.g., COL4A2) involved in cell adhesion (Rhodes *et al*, 2015) were upregulated. Data illustrated in Fig 2 suggest that elevated miR-483-3p/-5p inhibits PAH-related genes and pathways in the pulmonary endothelium.

## miR-483 targets TGF-β, TGFBR2, IL-1β, and ET-1 mRNAs

We then used RNAhybrid software to predict miR-483-targeted mRNAs. As illustrated in Fig 3A, while miR-483-3p was predicted to target the 3′UTR of TGF-β, TGFβR2, Smad2, ROCK1, β-catenin, and ET-1 mRNAs, miR-483-5p targets the 3′UTR of TGF-β, TGFBR2, Smad2, Smad3, IL-1β, and ET-1 mRNAs. Additionally, miR-483 targeting 3′UTR of TGF-β, TGFBR2, IL-1β, and ET-1 mRNAs was found conserved among human, rat, and mouse (Appendix Table S1). To validate the *in silico* predictions, pre-miR-483 was overexpressed in human PAECs by lentivirus (Lenti-miR-483, Fig 3B). As predicted, the mRNA and protein levels of TGF-β, TGFBR2, β-catenin, CTGF, IL-1β, and ET-1 were lower in PAECs infected with Lenti-miR-483 (Fig 3C and D). In the complementary approach, miR-483 inhibition by anti-miR-483 resulted in increased mRNA and protein levels of TGF-β, TGFBR2, β-catenin, CTGF, IL-1β, and ET-1 (Appendix Fig S3A–C). We then examined whether miR-483-3p/-5p directly targets these genes

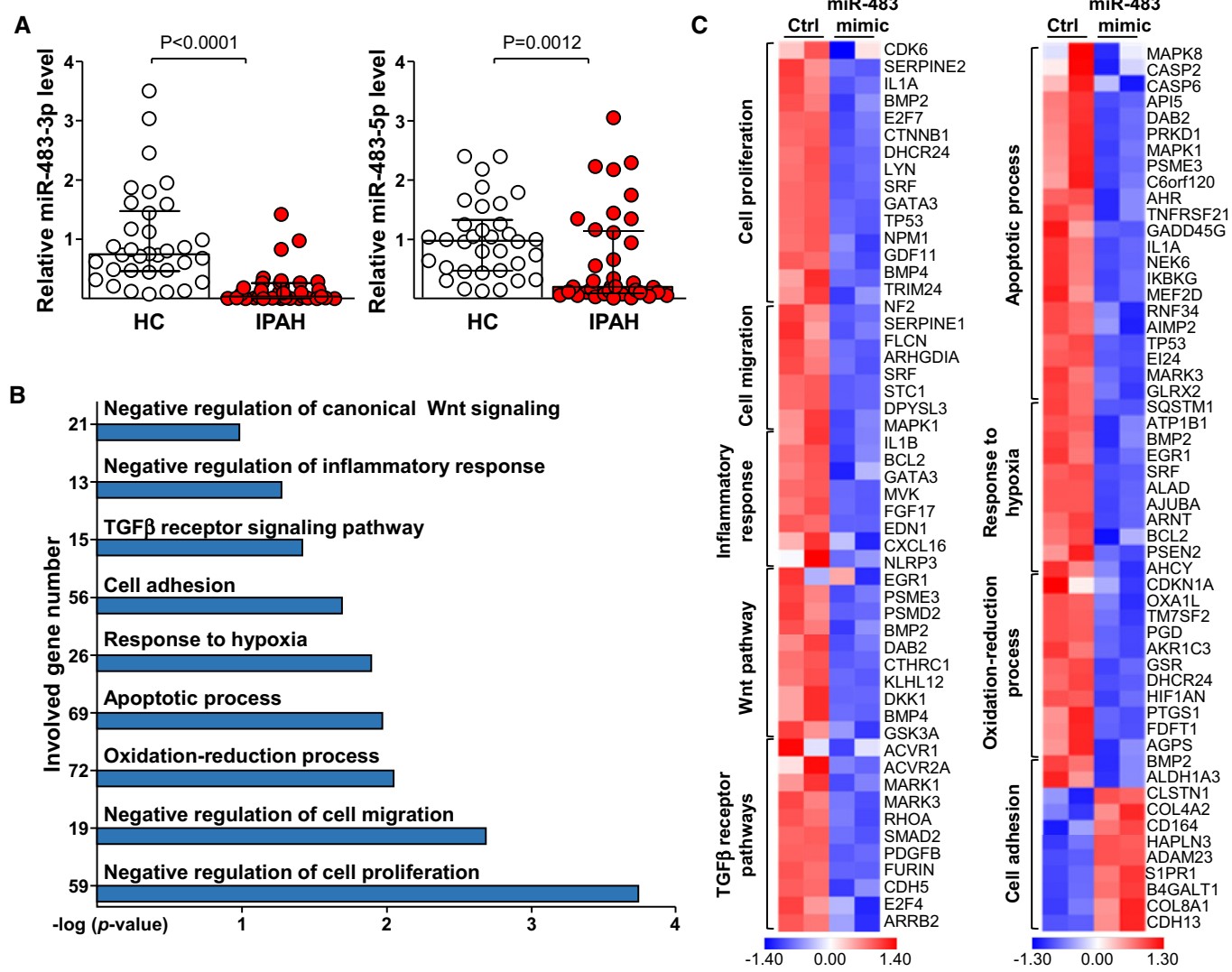

**Figure 2. MiR-483 targets PAH-related genes and pathways in PAECs.**

A CD144-enriched EVs were isolated from serum of IPAH patients (n = 37) and HCs (n = 34). The levels of miR-483-3p/-5p were measured by qPCR.

B, C PAECs were transfected with miR-483-3p/-5p mimic or scramble RNA for 36 hr before RNA isolation and then analyzed by RNA-seq. Data are results from two biological repeats. (B) PAH-related GO enrichment delineated by DAVID for the top 300 upregulated or downregulated genes with the cutoff of P < 0.05. (C) Heat map comparison of log2 fold changes of the indicated genes.

Data information: Values are expressed as median ± interquartile range. Statistical test: Mann–Whitney U-test.

Source data are available online for this figure.

by using TGF-β, TGFBR2, IL-1β, and ET-1 3'UTR constructs conjugated with a luciferase reporter. As shown in Fig 3E, miR-483 overexpression decreased the luciferase activity of the wild-type reporters Luc-TGF-β (WT), Luc-TGFBR2 (WT), Luc-IL-1β (WT), and Luc-ET-1 (WT). However, no reduction of luciferase activity was observed in ECs transfected with Luc-TGF-β (mut), Luc-TGFBR2 (mut), Luc-IL-1β (mut), or Luc-ET-1 (mut) in which the miR-483-targeted sequences were mutated. Moreover, miR-483-3p/-5p levels were increased in the miR-induced silencing complexes (miRISC) (i.e., Ago1, Ago2) in PAECs overexpressing miR-483 (Fig 3F). Additionally, levels of TGF-β, TGFBR2, β-catenin, CTGF, IL-1β, and ET-1 mRNAs were also increased in association with Ago1 and Ago2 (Fig 3G), indicating these mRNAs were targeted by miR-483 in the miRISC.

**PH amelioration in EC-miR-483-Tg rats**

With the findings of reduced miR-483 level in circulation and PAECs of IPAH patients and that miR-483 targeted several PAH-related genes in cultured PAECs, we investigated whether a supraphysiological level of miR-483 mitigates experimental PH in rat models by generating a transgenic rat line with miR-483 expression driven by the VE-cadherin promoter (i.e., EC-miR-483-Tg). As expected, levels of miR-483-3p/-5p were higher in lung ECs from EC-miR-483-Tg rats than wild-type littermates (Fig 4A). A comparison of miR-483-3p/-5p levels in aortic ECs and peripheral blood mononuclear cells confirmed the EC-specific miR-483 overexpression (Appendix Fig S4). Consistently, miR-483 levels were elevated in CD144-enriched EVs

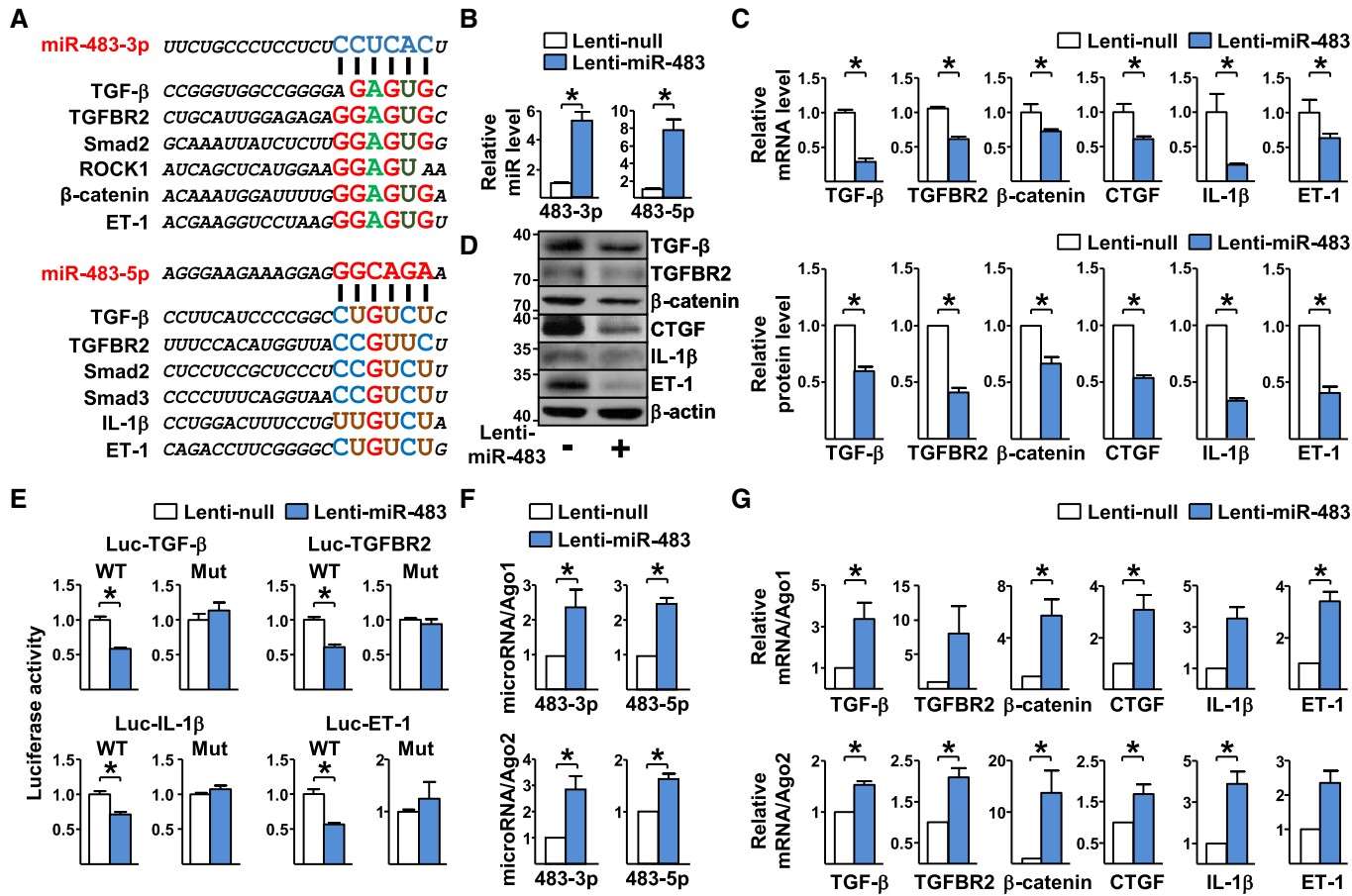

**Figure 3. MiR-483-targeted genes.**

A  Predicted binding sites for miR-483-3p/-5p on the 3′UTR of mRNAs as indicated.

B–D  PAECs were infected with Lenti-pre-miR-483 or Lenti-null for 24 hr. Expression levels of miR-483-3p/-5p, TGF-β, TGFBR2, β-catenin, CTGF, IL-1β, and ET-1 mRNA and protein were measured by qPCR and Western blot, respectively.

E  Bovine aortic ECs were transfected with a luciferase reporter fused with the 3′UTR of TGF-β (Luc-TGF-β WT), TGFBR2 (Luc-TGFBR2 WT), IL-1β (Luc-IL-1β WT), or ET-1 (Luc-ET-1 WT) or a binding site mutation (Luc-TGF-β Mut, Luc-TGFBR2 Mut, Luc-IL-1β Mut, ET-1-Mut), then infected with Lenti-pre-miR-483 for additional 36 hr. Luciferase activity was measured.

F, G  PAECs were infected with Lenti-pre-miR-483 or Lenti-null for 36 h. The Ago1- or Ago2-associated miRNAs and mRNAs were enriched by immunoprecipitation with anti-Ago1 or anti-Ago2. Levels of miR-483-3p/-5p and TGF-β, TGFBR2, β-catenin, CTGF, IL-1β, and ET-1 mRNA were detected by qPCR and normalized to those of Ago1 or Ago2 protein.

Data information: Values are expressed as mean ± SEM from three independent experiments. Statistical test: t-test (*P < 0.05 between the indicated groups, exact P-values are shown in Appendix Table S3).

Source data are available online for this figure.

in serum collected from EC-miR-483-Tg rats (Fig 4B), which suggests that these miRNAs were derived from endothelium.

We cultured lung ECs isolated from EC-miR-483-Tg and control wild-type rats under hypoxic conditions. Hypoxia increased the proliferation and migration of ECs from wild-type ECs but not those from EC-miR-483-Tg rats (Fig 4C and D). The basal protein and mRNA levels of TGF-β, TGFBR2, β-catenin, CTGF, IL-1β, and ET-1 in the lung homogenates in EC-miR-483-Tg rats were similar to or slightly lower than that in their wild-type littermates (WT+saline vs. Tg+saline in Fig 4E). MCT administration induced these PAH-related genes to much higher levels in the wild-type than those in EC-miR-483-Tg rats (WT+MCT vs. Tg+MCT in Fig 4E). Interestingly, circulation and lung levels of miR-483-3p/-5p were lower in wild-type rats receiving MCT, when compared to EC-miR-483-Tg rats receiving

MCT (Fig 4F). For miR-483 targeting, Ago1 and Ago2 miRISC complexes isolated from the lung homogenates of EC-miR-483-Tg rats treated with or without MCT were associated with increased miR-483-3p, miR-483-5p, along with TGF-β, TGFBR2, β-catenin, CTGF, IL-1β, and ET-1 mRNAs (Fig 4G and H). The PH-associated changes in mean PAP (mPAP) and the Fulton index (right ventricle weight to left ventricle plus interventricular septum weight, i.e., RV/LV+S) were ameliorated in EC-miR-483-Tg rats given MCT (Fig 5A and B). Vascular thickening and lumen closure seen in the wild-type rats receiving MCT were mitigated in EC-miR-483-Tg littermates with MCT administration (Fig 5C). Additionally, Evans blue dye staining showed increased permeability of endothelium in MCT-treated wild-type lungs, which was less severe in EC-miR-483-Tg lungs with MCT treatment (Fig 5D). For loss-of-function

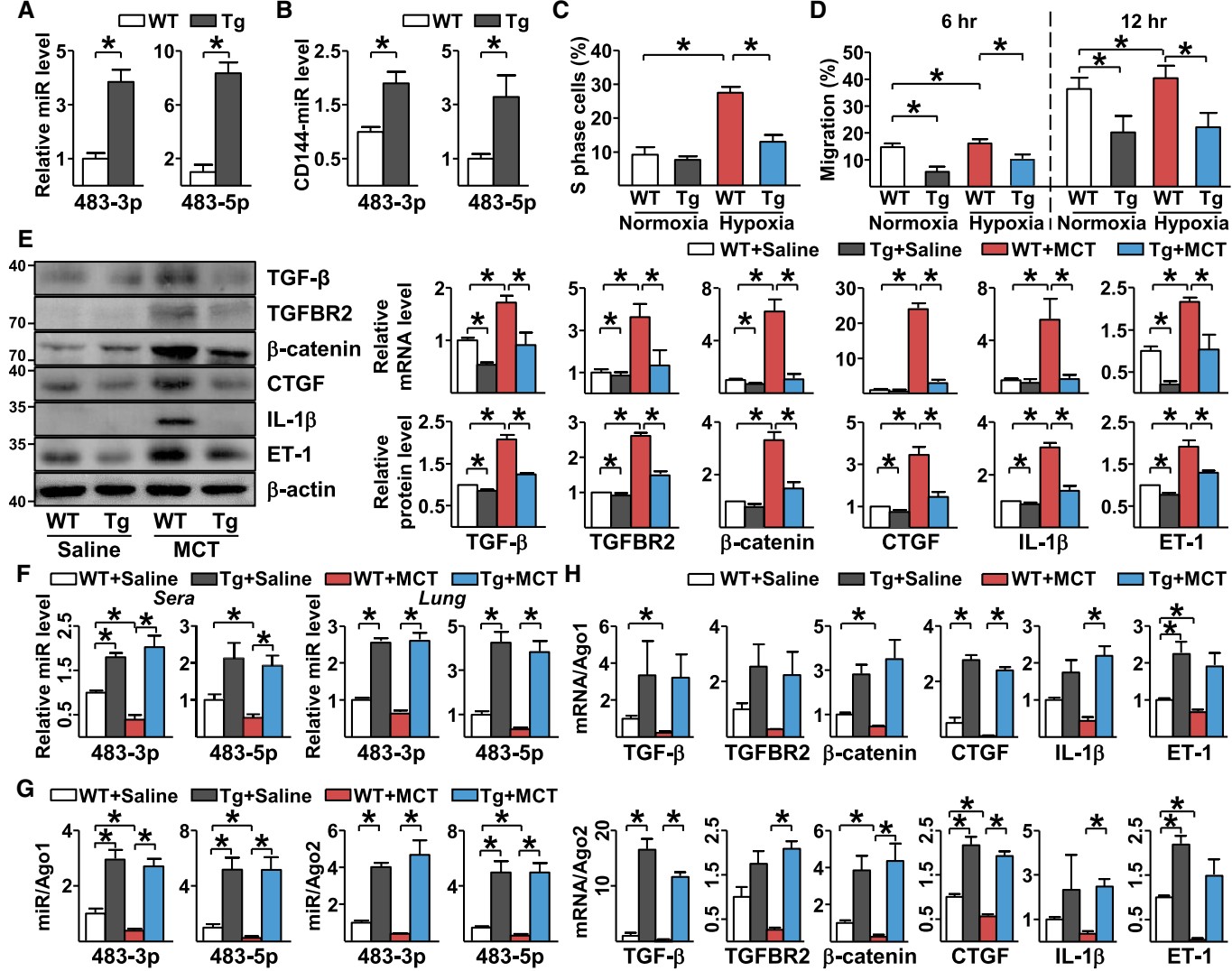

Figure 4. Decreased level of PAH-induced genes in EC-miR-483 Tg rats.

A    Lung ECs were isolated from EC-miR-483 transgenic rats (Tg) and their wild-type littermates (WT), n = 3 in each group. MiR-483-3p/-5p levels were measured by qPCR.

B    Serum from WT and Tg rats (n = 6) was collected. MiR-483-3p and miR-483-5p associated with CD144 were enriched by immunoprecipitation with anti-CD144. Levels of miR-483-3p/-5p were measured by qPCR.

C, D  Lung ECs isolated from miR-483-Tg rat and WT littermates were exposed to normoxia or hypoxia (0.2% $O_2$) for 24 hr. Proliferation and migration of ECs were measured by flow cytometry and wound healing assay, respectively.

E–H  Tg and WT rats were injected with saline or MCT. Expression levels of TGF-β, TGFBR2, β-catenin, CTGF, IL-1β, and ET-1 mRNA and protein in lung tissues were measured by qPCR and Western blot, respectively (E). Levels of miR-483-3p/-5p in serum and lung tissues were measured by qPCR (F). Ago1- or Ago2-associated miRNAs (G) and mRNAs (H) were enriched from the isolated lung tissues by immunoprecipitation with anti-Ago1 or anti-Ago2 and quantified by qPCR (n = 3×3, samples were pooled from three animals for each assay, and 3 independent experiments [a total of 9 animals] were performed).

Data information: Values are expressed as mean ± SEM. Data are representative of three independent experiments (C, D). Statistical test: t-test (*P < 0.05 vs. respective controls or between the indicated groups, exact P-values are shown in Appendix Table S3).

Source data are available online for this figure.

experiments, we administered locked nucleic acid (LNA)-modified antisense miR-483 (LNA-483) to MCT-treated rats. Compared with control rats receiving scramble RNA, the levels of miR-483 in sera and lung were decreased, while the mRNA and protein levels of PAH-related genes (e.g., TGF-β, TGFBR2, β-catenin, CTGF, IL-1β, and ET-1) were upregulated in rats having LNA-483 (Appendix Fig S5A–C). These results suggest that miR-483 could downregulate

PAH-related genes in experimental PH rats. With respect to phenotypic changes, mPAP, Fulton index, and pulmonary arterial remodeling were also exacerbated by LNA-anti-483 administration (Appendix Fig S5D–F).

The beneficial effect of miR-483 on mitigating experimental PH was also demonstrated in the SU5416/hypoxia-induced PH model. The onset of PH, as revealed by increased mPAP and Fulton index,

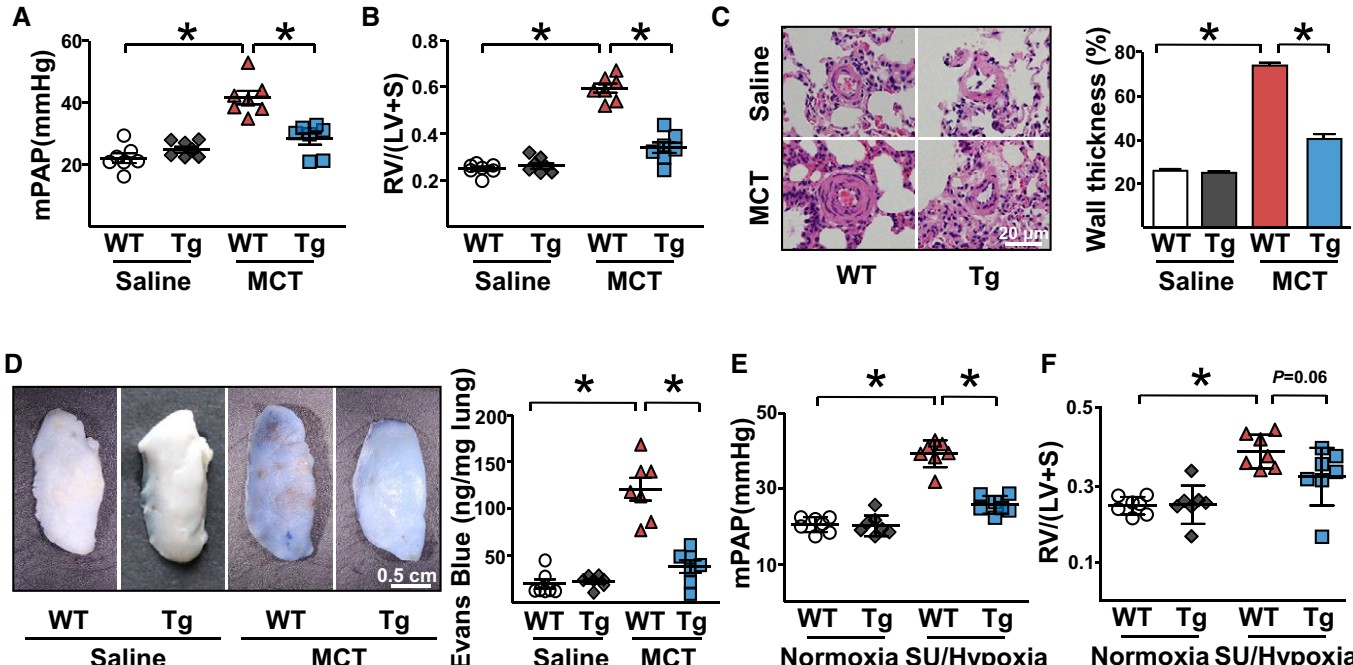

**Figure 5. EC-miR-483 Tg rats show ameliorated PH.**

A–D EC-miR-483-Tg and WT rats were injected with saline or MCT. Analysis of mPAP and RV hypertrophy [RV/(LV + S)] for the indicated groups (7 rats per group) (A, B). Vascularization evaluated by H&E staining of pulmonary arteries (C). Vascular integrity evaluated by intravenous injection of Evans blue dye (D).

E, F EC-miR-483-Tg rats and WT littermates were injected with SU5416 and then exposed to hypoxia for 3 weeks and had reoxygenation for 2 weeks or injected with DMSO and exposed to normoxia for 5 weeks. Analysis of mPAP (E) and RV hypertrophy [RV/(LV + S)] (F) for the indicated groups.

Data information: Values are expressed as mean ± SEM. Statistical test: t-test (*P < 0.05 vs. respective controls or between the indicated groups). Scale bar: 20 µm (C) or 0.5 cm (D).

Source data are available online for this figure.

tended to be attenuated in EC-miR-483-Tg rats versus wild-type rats, both with SU5416/hypoxia treatment (Fig 5E and F). Consistently, circulation and lung levels of miR-483 were reduced (Appendix Fig S6A), while TGF-β, TGFBR2, β-catenin, CTGF, IL-1β, and ET-1 mRNAs and protein levels in lung tissues were increased in Su5416/hypoxia-treated WT rats but not in EC-miR-483-Tg rats (Appendix Fig S6B and C).

### Intratracheal administration of miR-483 reduces PH susceptibility

To test the efficacy of exogenously administered miR-483 to alleviate PH, we delivered miR-483 to rats via inhalation of Lenti-miR-483-GFP. Following intratracheal administration (Fig 6A), the circulatory and pulmonary levels of miR-483-3p/-5p were significantly higher in rats receiving Lenti-miR-483-GFP versus Lenti-GFP (Fig 6B and C). The efficacy of the lentivirus administration was further demonstrated by flow cytometry showing the presence of GFP in the lungs of rats that received the lentivirus (Appendix Fig S7). Lenti-miR-483-GFP administration attenuated the MCT induction of TGF-β, TGFBR2, β-catenin, CTGF, IL-1β, and ET-1 (Fig 6D) and reversed the MCT-induced PH (Fig 6E and F). Neither MCT nor Lenti-miR-483-GFP affected systemic blood pressure in the 3 groups (Fig 6G). The protective effect of miR-483 in attenuating PH is also supported by comparing RV enlargement (i.e., RV/LV, Fig 6H) and vessel thickness and occlusion (Fig 6I). In line with the mitigated

PH, Lenti-miR-483-GFP administration seemed to improve the survival of MCT-administered rats (Fig 6J). Overall, the results in Fig 6 suggest that exogenously administered miR-483 reversed MCT-induced PH in rat models.

## Discussion

Data from IPAH patients, PH rat models, and cultured PAECs provide evidence that low miR-483 is associated with propensity to disease and that elevated levels are protective. We show that miR-483 targets the 3′-UTR of multiple genes implicated in PAH, including those involved in TGF-β signaling (e.g., TGF-β, TGFBR2), inflammation (e.g., IL-1β), and vasoconstriction (e.g., ET-1). Given PAH risk stratification is critical for initial therapy strategy and prognosis evaluation (Galie et al, 2019), here we show that lower circulating levels of miR-483 among IPAH patients had worse prognosis (Fig 1C). Thus, the novelty of our observations is that miR-483 provides a unifying factor that can address multiple adverse responses associated with PAH.

Lower levels of miR-483 in lung ECs isolated from PH rats (Appendix Fig S8) were coincident with increased expression of PAH-related genes (e.g., ET-1). Thus, an optimal level of miR-483 in pulmonary endothelium may mitigate PAH-related EC dysfunction. Indeed, miR-483 overexpression in cultured PAECs induced the expression of genes critical for EC homeostasis that include

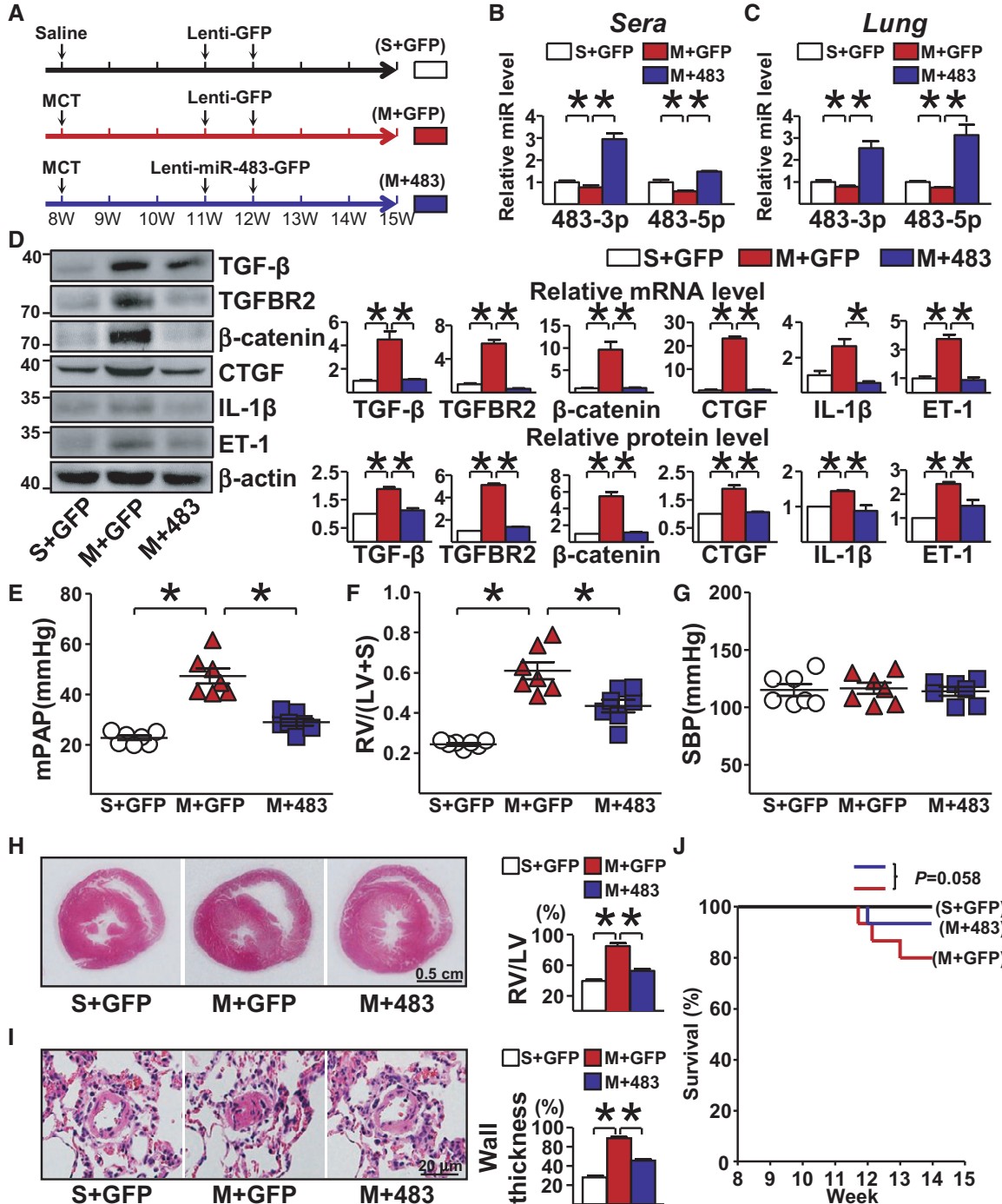

**Figure 6. Lenti-miR-483 inhalation ameliorates MCT-induced PAH.**

A   Ctrl- or MCT-treated rats were intratracheally delivered Lenti-pre-miR-483-GFP or Lenti-GFP at Day 21 and Day 28 after the start of MCT treatment (S+GFP, M+GFP, and M+483).

B, C   qPCR analysis of miR-483-3p/-5p in serum and lung tissue from PH (MCT-treated) rats.

D   Protein and mRNA levels of TGF-β, TGFBR2, β-catenin, CTGF, IL-1β, and ET-1 measured by Western blot and qPCR, respectively (n = 3×3, samples were pooled from three animals for each assay, and 3 independent experiments [a total of 9 animals] were performed).

E–I   Analysis of mPAP (E), RV hypertrophy [RV/(LV + S)] (F), and systemic blood pressure (SBP) (G) for the indicated groups (7 rats in each group). (H) RV hypertrophy shown by the H&E staining and RV/(LV + S) quantification. (I) Vascularization revealed by H&E staining (n = 7 rats in each group).

J   Survival rate of three groups (n = 15 rats in each group).

Data information: Values are expressed as mean ± SEM. Statistical test: t-test (*P < 0.05 vs. respective controls or between the indicated groups, exact P-values are shown in Appendix Table S3). Scale bar: 0.5 cm (H) or 20 μm (I).

Source data are available online for this figure.

Krüppel-like factor 2 (KLF2), KLF4, eNOS, ACE2, and APLNR (Appendix Fig S9; Shen *et al*, 2009; Alastalo *et al*, 2011; Huang *et al*, 2017; Zhang *et al*, 2018). Interestingly, KLF4 overexpression, statins treatment, and pulsatile shear stress, known to enhance endothelial function, also increase the expression of miR-483 in ECs (He *et al*, 2017; Fernandez Esmerats *et al*, 2019). As an intronic miRNA, miR-483 is co-expressed with IGF2. However, IGF2 has not been implicated in PAH, unlike other growth factors such as TGF-β, VEGF, IGF-1, and PDGF. Thus, the mechanism underlying PAH attenuation of miR-483 via *IGF2* in pulmonary endothelium warrants further study.

Levels of miR-483-3p/-5p were reduced in the serum, lung ECs, and CD144-enriched EVs from IPAH patients and PH rats (Fig 2A, Appendix Fig S10). Thus, decreased miR-483 in CD144-enriched EVs may be due to decreased miR-483 expression in the damaged endothelium. It can also be expected that EC-derived miRNAs are released into the subendothelium and taken up by vascular smooth muscle cells (VSMCs; Zhu *et al*, 2017). The level of miR-483 was indeed increased in the tunica media of the pulmonary artery in EC-miR-483-Tg rats and in pulmonary arterial SMCs co-cultured with lung ECs isolated from EC-miR-483-Tg rats (Appendix Fig S11), suggesting that EC-derived miR-483 suppresses SMC proliferation and inflammation.

Besides miR-483, other miRNAs have been implicated in PAH (Kim *et al*, 2013; Rothman *et al*, 2016). Among reported miRNAs involved in PAH, miR-483 is significant because miR-483 level was inversely correlated with the severity of clinical manifestations and miR-483 targets multiple genes involved in PAH. Other than TGF-β, TGFBR2, β-catenin, CTGF, IL-1β, and ET-1 mRNA, hsa-miR-483-3p is likely to target Smad2, ROCK1, IL-6, IGF1R, MMP9, NOTCH3, and PP2A, whereas hsa-miR-483-5p targets Smad2, Smad3, IL-6, NOTCH3, and ACVR1 (Appendix Fig S12). Moreover, many genes downregulated by miR-483, shown in Fig 2D, do contain miR-483-3p/-5p target sites in their 3′-UTR (Appendix Fig S13). To our knowledge, none of the reported miRNA can target PAH as comprehensively as miR-483.

Data illustrated in Fig 1 reveal that serum levels of miR-483-3p better correlate with clinical features of PAH, when compared to miR-483-5p. During miRNA processing, the mature miRNA strand of the miRNA:miRNA* duplex is selectively incorporated into the miRISC for mRNA targeting, whereas the miRNA* strand degrades due to its exclusion from the miRISC complex (Khvorova *et al*, 2003). It is likely that miR-483-3p is the mature strand and miR-483-5p the miRNA* strand. This thesis is supported by that qPCR Ct values for miR-483-3p levels in ECs, human and rat serum, and miRISC (i.e., Ago1, Ago2) were several cycles lower than those of miR-483-5p (Appendix Fig S14). Hence, miR-483-3p might be more therapeutically effective.

Clinically, epoprostenol, PDE5 inhibitors, and endothelin receptor antagonists are limited in their efficacy as phenotypic modulators of altered vascular function. In this study, intratracheal delivery of miR-483 at 3 weeks after MCT administration reduced mPAP, RV hypertrophy, and vascular remodeling (Fig 6). In line with this finding, the supraphysiological level of miR-483 in EC-miR-483-Tg rats led to a PH-resistant phenotype (Fig 5). Our findings herein demonstrate the potential use of exogenously administered miR-483 or agents that elevate endogenous miR-483 to maintain functional endothelium, at least during the early stages of PAH.

# Materials and Methods

### Human subjects and serum sampling

In total, 139 IPAH patients were prospectively enrolled from June 2015 to February 2018 at FuWai Hospital, Chinese Academy of Medical Sciences. Patients diagnosed with IPAH met the published guidelines for the diagnosis of pulmonary hypertension (Galie *et al*, 2016a,b), i.e., mPAP ≥ 25 mmHg at rest, pulmonary artery wedge pressure ≤ 15 mmHg, and pulmonary vascular resistance (PVR) > 3 Wood units (Galie *et al*, 2008). Patients were excluded if associated with a definite cause, including connective tissue disease, congenital heart disease, chronic pulmonary thromboembolism, and PAH due to left heart disease, lung diseases, and hypoxemia. A group of 95 age- and sex-matched healthy subjects were recruited from FuWai Hospital as well. The investigation conformed to the principles outlined in the Declaration of Helsinki and the Department of Health and Human Services Belmont Report. The study protocol was approved by the ethics committees of FuWai Hospital, and written informed consent was obtained from all subjects. Blood samples were drawn from the cubital vein within 1 week from the time of right heart catheterization. Blood samples were taken from IPAH patients or healthy subjects in the fasting state for approximately 12 h. None of the individuals had exercise before blood collection. After centrifugation at 1,500 *g* for 10 min, the serum was aliquoted into separator tubes, quickly frozen in liquid nitrogen, and stored at −80°C until use. The experiments were approved by Ethics Committee of Xi'an Jiaotong University (No. 2018-544).

### Serum ET-1 and IL-6 levels

Serum ET-1 and IL-6 levels were measured by ELISA (Abcam, R&D Systems, respectively), according to the manufacturer's instructions. Briefly, serum samples were diluted 1:4 and then incubated in pre-coated plates. After 30-min incubation with horseradish peroxidase-labeled secondary antibody, TMB substrate solution was added. After another 30-min incubation, stop buffer was added and plates were read immediately by using a microplate reader set at 450 nm (Infinite M200 Pro, Tecan).

### CD144-enriched EVs

CD144-enriched EVs were isolated as described with modifications (Shang *et al*, 2017). Briefly, CD144-enriched EVs were immunoprecipitated from human or rodent serum (1 ml) by immunoprecipitation (IP) with anti-CD144 antibody (Santa Cruz Biotechnology) (1 µg per sample) incubating with Dynabeads (Invitrogen). IgG was used as an isotype control of IP. Total RNA from CD144-enriched EVs was isolated with TRIzol and with Caenorhabditis elegans miR-39 (Cel-miR-39) added at 2 nM as a spike-in control.

### Library preparation, RNA-sequencing, and data analysis

Pulmonary arterial endothelial cells were transfected with an equimolar of miR-483-3p and miR-483-5p mimic or control mimic (10 nM, Thermo Fisher Scientific) for 36 h, and total RNA was extracted with TRIzol (Invitrogen). RNA libraries were

prepared by using the NEBNext Ultra RNA Library Prep Kit (Illumina) following the manufacturer's protocols. The clustering of the index code samples was performed on a cBot Cluster Generation System with the TruSeq PE Cluster Kit v3-cBot-HS (Illumina) and sequenced on an Illumina HiSeq platform. Raw sequencing data (reads) in fastq format were processed with custom-made Perl scripts that removed reads containing adaptor poly-N and low-quality sequences. Clean data were then mapped to the hg19 whole genome by using Hisat2 v2.0.5 (Pertea *et al*, 2016). Feature Counts v1.5.0-p3 was used to count the reads numbers mapped to each gene (Liao *et al*, 2014). Then, reads per kilobase of exon per megabase of library size (RPKM) were calculated based on the length of the gene and reads count mapped to this gene, which assembles transcripts and their abundances, and estimates for gene expression levels. Differential expression analysis of two groups was performed by using the DESeq2 R package (Love *et al*, 2014). GO enrichment analysis was performed by DAVID to identify differentially regulated pathways between two groups (Jiao *et al*, 2012).

### EC-specific miR-483 transgenic rats

EC-miR-483-Tg rats were generated by the Nanjing Biomedical Research Institute of Nanjing University. Briefly, the mouse 2.5-kb Cdh5 promoter and rat full length of pre-miR-483 sequence were cloned by PCR from C57BL/6 mouse and Sprague-Dawley (SD) rat genomic DNA, respectively. The 0.4-kb rabbit beta globin polyA was derived from the plasmid pCAG-Cre-GFP. EC-miR-483-Tg founder rats were generated by pronuclear injection of fertilized eggs of outbred rats. Pups of EC-miR-483-Tg and their WT littermates were generated by crossing with SD rats. Eight- to 12-week-old male Tg rats and their age-matched WT littermates were used for all experiments.

### Rat PH models

Animal experiments were approved by the Institutional Animal Ethics Committee of Xi'an Jiaotong University (No. XJTULAC2014.212). SD rats were purchased from Xi'an Jiaotong University Experimental Animal Center. All rodent models were kept on a 12-h light/dark cycle and fed *ad libitum* with chow diet at temperature of 22°C. Male rats (300–350 g body weight) were administrated MCT (60 mg/kg; Sigma) or equal volume of saline by subcutaneous injection (Schermuly *et al*, 2005; Savai *et al*, 2014). At 21 or 28 days after MCT injection, rats were given Lenti-GFP or Lenti-miR-483-GFP orotracheally (Savai *et al*, 2014). Briefly, rats were anesthetized. Then, curved blunted-ended forceps were used to grasp the tongue to gain visualization of the larynx. The lentivirus ($1 \times 10^9$ PFU), diluted in 100 μl sterile PBS, was instilled using a PE-10 tubing inserted into the trachea. For Sugen/hypoxia plus reoxygenation-induced PH model, male rats (300–350 g) were subcutaneously injected with the vascular endothelial growth factor receptor inhibitor SU5416 (Sigma-Aldrich) at 20 mg/kg body weight and housed under hypoxic conditions (10% $O_2$) for 3 weeks. Rats were then exposed to normoxia for 2 weeks (reoxygenation). Rats receiving DMSO for the same durations were negative controls (Schermuly *et al*,

2005). Animals were randomized for interventions. Rats were euthanized by pentobarbital sodium before measurement of hemodynamics. For assessing RV hypertrophy, RV was dissected and weighed. We used picture scale plate to measure the thickness of pulmonary arteries (20–70 μm in diameters, $n = 10$ for each rat). The external diameter ($D_{ex}$) and inner diameter ($D_{in}$) of each artery were measured, and then, the wall thickness was assessed using ($D_{ex} - D_{in})/D_{ex} \times 100\%$. The ratio of right ventricle weight to left ventricle plus interventricular septum weight (RV/LV+S; i.e., Fulton index) was assessed. Researchers were blinding for measurement of hemodynamics, Fulton index, and wall thickness, but aware which intervention group corresponded to which cohort of animals when analyzed the data.

### Cell culture

Pulmonary arterial endothelial cell and the EGM-2 BulletKit medium were obtained from Lonza. Bovine aortic ECs (BAECs) and human embryonic kidney 293T (HEK293T) cells were obtained from ATCC. Cells were tested for *Mycoplasma* contamination. Rat pulmonary arterial smooth muscle cells (PASMCs) were isolated as described (Schermuly *et al*, 2005; Savai *et al*, 2014). PASMCs, BAECs, and HEK293T cells were cultured in Dulbecco's modified Eagle medium (DMEM) supplemented with 10% fetal bovine serum and 100 U/ml penicillin–streptomycin. PAECs with passages 4–7 were used for all cell culture experiments. All cells were cultured at 37°C in 5% $CO_2$ and 95% relative humidity.

### Rat lung EC isolation

Primary rat lung ECs were isolated as described (Lim & Luscinskas, 2006). Briefly, rat lungs were isolated from various groups of rats and perfused with 20 ml sterile PBS. The minced lung tissues were transferred to 50-ml tubes containing 15 ml 96% type I collagenase (Abcam) and incubated at 37°C with gentle agitation for 45 min. Cell suspensions were filtered through 70-μm cell strainers (Thermo Fisher Scientific) after 12 trituration cycles with 20-ml syringes. The cell suspension was washed and further precipitated with anti-CD31 (Abcam) pre-conjugated Dyna-beads (Invitrogen).

### Wound healing assay

Lung ECs were plated in 6-well plates. Wounds were created with a 200-μl pipette tip through the cells when they were about 90% confluence. Then, cells were cultured in normoxia or hypoxia (0.2% $O_2$). Images of cells were taken using microscopy (Olympus) at 0 ($W_0$), 6 ($W_6$), and 12 h ($W_{12}$) postwounding. The areas were randomly selected, and widths of wounded areas were measured. Migration was calculated by ($W_6 - W_0)/W_0 \times 100\%$ and ($W_{12} - W_0)/W_0 \times 100\%$.

### Luciferase reporter plasmids, transfection, and luciferase assay

The 3′ untranslated region (3′UTR) of human TGF-β, TGFBR2, IL-1β, and ET-1 gene containing the putative miR-483 targeting sites was synthesized by Shanghai Personal Biotechnology and subcloned into the pmirGLO dual-luciferase miRNA target expression vector (Promega) to obtain Luc-TGF-β-3′UTR-WT, Luc-TGFBR2-3′UTR-WT,

Luc-IL-1β-3′UTR-WT, and Luc-ET-1-3′UTR-WT reporter constructs. Mutant constructs were further created with "GTGATCGC" to "GGGGTGGC" replacement in Luc-TGF-β-3′UTR-mut; "AGAG-GAGTG" to "TAGTACTAT" in Luc-TGFBR2-3′UTR-mut; "CTGTTGTCT" to "CAGTAGACA" in Luc-IL-1β-3′UTR-mut; and "GGAGTG" to "GCAATC" in Luc-ET-1-3′UTR-mut. The various reporter constructs were transfected into BAECs by using Lipofectamine 2000 (Invitrogen). The luciferase activity was measured by using the Dual-Glo Luciferase Reporter Assay Kit (Promega) with a luciferase reader (PerkinElmer).

## Ago1-IP and Ago2-IP

Ago1-IP and Ago2-IP were performed as described (He *et al*, 2017). Briefly, PAECs or homogenized lung tissues were lysed in a buffer containing 50 mmol/l Tris–HCl pH 7.5, 150 mmol/l NaCl, 0.1% NP-40, 1 mmol/l EDTA, and 100 U/μl RNase inhibitor. An amount of 1 ml lysates was incubated overnight with 100 μl protein G Dynabeads conjugated with 5 μl anti-Ago1 (Wako Chemicals) or anti-Ago2 (Cell Signaling Technology) at 4°C. The immunoprecipitated RNAs were then extracted with TRIzol. A complete list of primers used for Ago1-IP and Ago2-IP is in Appendix Table S2.

## RT–qPCR and Western blot analysis

RNA was extracted from cultured cells by using TRIzol. For serum samples, RNA was isolated by using TRIzol LS from 200 μl serum. Cel-miR-39 was added at 2 nM as a spike-in control. Relative mRNA or miRNA expression was determined by using SYBR Green (Bio-Rad) or TaqMan probe-participated qPCRs, GAPDH and U6 served as internal controls for normalization of mRNA and miRNA expression, respectively. The sequences for qPCR primers are listed in Appendix Table S2. Proteins were isolated by using RIPA lysis buffer and separated by SDS–PAGE. After being transferred to PVDF membranes, membranes were incubated with various antibodies as indicated. Protein bands were detected by electrochemiluminescence with horseradish peroxidase-labeled secondary antibodies. The used primary antibodies were as follows: TGF-β (Abcam, ab92486, 1:1,000); TGFBR2 (Santa Cruz, sc-17792, 1:500); β-catenin (Abcam, ab32572, 1:1,000); CTGF (Santa Cruz, sc-365970, 1:1,000); IL-1β (Abcam, ab2105, 1:1,000); and ET-1 (Abcam, ab2786, 1:1,000). Secondary antibodies used were as follows: anti-mouse (Jackson, 515-035-003, 1:5,000) and anti-rabbit (Jackson, 111-035-045, 1:5,000).

## Evans blue staining

An amount of 2 ml of Evans blue solution (0.5% W/V in PBS) was injected into rat tail veins. After 30 min, rats were euthanized and perfused with 50 ml PBS to remove intravascular dye. The left lung was fixed using 4% paraformaldehyde, and images were taken. Four milliliters of formamide (Fisher Scientific) was added to the homogenized right lung tissues and incubated at 55°C for 48 h to extract Evans blue from the tissues. The absorbance at 620 nm of 200 μl clarified supernatant aliquots was measured spectrophotometrically (PerkinElmer VICTOR ×2). Quantification of Evans blue staining was calculated by measuring the nanogram dye extracted per milligram tissue.

### The paper explained

#### Problem

PH is characterized by increased pulmonary arterial pressure and small pulmonary vascular remodeling, resulting in right ventricular hypertrophy and finally heart failure. Mounting evidence indicates that endothelial dysfunction plays an important role in regulating the pulmonary arterial remodeling during the pathogenic process of PH. The current clinical therapy for PH mainly focuses on the dilation of pulmonary vascular by targeting pathways related to ET-1, NO, and prostaglandins. However, these treatments are not curative. Micro-RNAs are small molecules that target mRNA transcripts and inhibit their expression, which are implicated in EC health and disease. The current study explores whether miR-483, a protective miR in ECs, could protect against PH pathogenesis.

#### Results

Our results showed that miR-483 was decreased in the sera of patients with IPAH, especially those with more severe disease. RNA-sequencing and bioinformatics analyses revealed that miR-483 decreases and targets a panel of PAH-related genes, including TGF-β, TGFBR2, β-catenin, CTGF, IL-1β, and ET-1. Overexpression of miR-483 in PAECs inhibits these PAH-related genes. When challenged with MCT or Sugen + hypoxia, rats with EC-specific miR-483 overexpression exhibited ameliorated PH and reduced right ventricular hypertrophy. Moreover, exogenously delivered miR-483 suppressed PAH-related gene expression and reversed the PH pathogenesis in rats received MCT.

#### Impact

Our findings suggest that decreased circulating level of miR-483 is associated with human IPAH subjects and rodent PH models. Overexpression of miR-483 suppresses the PAH-related genes and therefore tends to protect EC function and ameliorate experimental PH.

## Statistical analysis

Experimental intervention was randomized. Researchers were blinded during measurement of serum miR-483 levels, hemodynamics, Fulton index, and wall thickness. The distribution of quantitative variables was assessed for normality by using 1-sample Kolmogorov–Smirnov test. Equal variance assumptions were tested with the Brown–Forsythe test. Because serum levels of miR483-3p/-5p were highly skewed, the difference between IPAH and control subjects was compared by Kruskal–Wallis *U*-test or Mann–Whitney *U*-test. Other quantitative variables were evaluated with Student's *t*-test or one-way ANOVA. Multiple group testing was adjusted with the Bonferroni correction. Results are reported as percentages, median (interquartile range), or means ± SEM, as indicated. Two-tailed $P < 0.05$ was considered statistically significant. All *P*-values are listed in Appendix Table S3. All analyses were performed with PASW Statistics 18.0 (SPSS Inc).

## Data availability

The datasets produced in this study are available in the following databases:

RNA-Seq data: Gene Expression Omnibus GSE146774 (https://www.ncbi.nlm.nih.gov/geo/query/acc.cgi?acc = GSE146774).

**Expanded View** for this article is available online.

## Acknowledgements

The authors would like to thank Dr. Yan Wu at FuWai Hospital, Drs. Baochang Lai and Ting Lei, and Ms. Jie Li in XJTU for their technical assistance. This work was supported by the Beijing Natural Science Foundation (7181009), 13th Five-Year Plan—Precise Medicine—Key Research and Development Program—Clinical Cohort of Rare Disease (2016YFC0901500), Key Project of Natural Science Foundation of China (81630003), CAMS Innovation Fund for Medical Sciences (2016-I2M-1-002, 2017-I2M-B&R-02), National Natural Science Foundation of China (81270349, 81670452), Beijing Nova Programme Interdisciplinary Cooperation Project (XXJC201805-Z181100006218125), and NIH research grants K24 HL132105 and K99 HL135258.

## Author contributions

JZ performed and analyzed PAEC experiments, sera experiments, and most animal experiments; YH performed sera experiments and clinical analysis; XY, SC, ZL, and YL performed animal experiments and some PAEC experiments. MH, YS, and LB helped with the statistical analysis. MH, JZ, YM, MG, XC, XW, YZ, AM, and FF helped with the clinical analysis; MH, JZ, C-HC, H-DH, and BG provided RNA-seq analysis; MR and Z-CJ provided human samples from IPAH patients. Z-CJ and JY-JS designed and supervised the research and wrote the manuscript.

## Conflict of interest

The authors declare that they have no conflict of interest.

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
