## [Review Process File · EMBO Molecular Medicine]

MicroRNA-483 amelioration of experimental pulmonary hypertension

Jin Zhang, Yang-Yang He, Xiaosong Yan, Shanshan Chen, Ming He, Yuyang Lei, Jiao Zhang, Brendan Gongol, Mingxia Gu, Yifei Miao, Liang Bai, Xiaopei Cui, Xiao-Jian Wang, Yixin Zhang, Fenling Fan, Zhao Li, Yuan Shen, Chih-Hung Chou, Hsien-Da Huang, Atul Malhotra, Marlene Rabinovitch, Zhi-Cheng Jing and John Shyy

Review timeline:

Submission date:	16 th August 2019
Editorial Decision:	10 th September 2019
Revision received:	16 th December 2019
Editorial Decision:	14 th January 2020
Revision received:	20 th February 2020
Editorial Decision:	9 th March 2020
Revision received:	13 th March 2020
Accept:	17 th March 2020

Editor: Jingyi Hou

Transaction Report:

1st Editorial Decision

10th September 2019

Thank you for the submission of your manuscript to EMBO Molecular Medicine. We have now received feedback from the three reviewers who agreed to evaluate your manuscript. As you will see from the comments below, while referee #3 is more supportive of publication, referees #1 and #2 raise a number of concerns with regard to the link between miR-483 and human PAH, technical quality, missing details, and the lack of loss-of-function studies on miR-483.

Upon our cross-commenting exercise, referee#2 commented "(i) technical improvements and (ii) in vivo loss-of-function analysis of miR-483 are minimum essential for consideration for publication in EMBO Molecular Medicine." Referee #3 added "In rats with MCT-PH, I would encourage the authors to conduct a pharmacological experiment using miR-483 antagmir to see whether inhibition of miR-483 has protective effect (on, e.g., pulmonary hemodynamics/PAP/RVSP, Fulton Index for RH hypertrophy, lung vascular histological and morphological changes...the routine measurements for in vivo animal pharmacological experiments)."

We would expect a point-by-point response to all concerns raised by the referees, along with providing additional details and clarifications. In particular, in vivo loss-of-function analysis of miR-483 needs to be performed as requested by referee #2 to improve the conclusiveness of the study. We would also encourage you to perform a pharmacological inhibition experiment as recommended by referee #3.

***** Reviewer's comments *****

Referee #1 (Comments on Novelty/Model System for Author):

Zhang et al report a novel role for miR-483 in the pathobiology of pulmonary arterial hypertension. This link between miR-483 and PAH appears new, though many other miRNAs since 2010 have been implicated in PAH. The main strength of this study is the diversity of techniques and specimens that were used, including animal experiments that suggest potential therapeutic effects of miR-483. However, there are several major criticisms. First, the link between miR-483 and human PAH is rather weak. The lower levels of miR-483 in IPAH versus healthy control serum is extremely modest, and could very well be caused by technical artifacts related to differential quality of the serum or extracted RNA (which is not addressed in the manuscript). The reported normalization controls cannot compensate for this potential issue. Moreover, miR-483 levels in PAECs from IPAH patients showed no significant differences compared to healthy controls. Second, there is a general lack of transparency and/or clarity in the reporting of methodological details (even after reviewing the data supplement), which reduces the credibility of the results. The sample sizes for many experiments are very small (as low as n=1 per group in some cases), which also limits the robustness of the conclusions, and it's often not even clear what sample size was used for a particular experiment. Finally, there are a number of instances where the authors have overstated their results (i.e., where the data simply does not support the conclusion), which are outlined below in addition to other specific comments and questions.

Referee #1 (Remarks for Author):

Zhang et al report a novel role for miR-483 in the pathobiology of pulmonary arterial hypertension. This link between miR-483 and PAH appears new, though many other miRNAs since 2010 have been implicated in PAH. The main strength of this study is the diversity of techniques and specimens that were used, including animal experiments that suggest potential therapeutic effects of miR-483. However, there are several major criticisms which greatly limit the impact of this work as detailed below.

Specific comments:

1. When were the serum samples collected and over what time period? This information should be reported in the methods to provide some insight into the age-related quality of the samples. The information should be reported separately by IPAH patients and healthy controls to rule out the potential for underlying technical artifacts.
2. Please report the passage number(s) related to cell culture experiments.
3. At what level is miR-483-3p/-5p present in serum? Please report Cq values to provide some indication of the detection level in serum.
4. Please clarify how miR-483 levels in plasma were normalized? Was the data normalized to cel-miR-39 or U6. If normalized to U6, please report the detection level of U6 in the serum.
5. Please report the volume of serum per patient used for extraction.
6. Please report how much RNA per sample was used for reverse transcription and how was this standardized across samples?
7. What evidence can the authors provide to indicate that the quantity and quality of serum-extracted RNA was comparable between samples (especially between healthy and IPAH groups)? This is an important consideration since cel-miR-39 is a poor control (as compared to an internal reference control) since it only accounts for variations incurred after the spike-in event. How do the authors know the RNA in the IPAH serum samples was not degraded, leading to reduced miRNA levels?
8. The difference in serum miR-483 levels between IPAH and healthy controls is very modest, suggesting it would be a poor biomarker. Moreover, there is little value in a biomarker that can distinguish between IPAH and healthy people (as opposed to more clinically relevant control groups).

9. Results Pg 7: The author's statement that, "Patients with higher mPAP and PVR seemed to have lower miR-483-3p/-5p levels (Supplementary Fig. 1A, B)." is not supported by the data in supplementary Fig 1A and B. I don't see any consistent and clear differences between the stratified groups, so the data is at best inconclusive.
10. Results Pg 7: The author's statement that, "...the 3 determinants of prognosis according to 2015 ESC/ERC guidelines for PH [mean right atrial pressure (mRAP), Cardiac index (CI), WHO functional class, and N-terminal pro-brain natriuretic peptide (NT-proBNP)] were also related to miR-483-3p/-5p levels (Supplementary Fig. 1C-F)," is again not supported by the data in the supplementary figures. There doesn't appear to be any differential response related to clinical severity of PAH.
11. Results Pg 7: Regarding the author's statement, " Interestingly, the serum levels of ET-1 and IL-6 were inversely correlated with those of miR-483-3p (Fig. 1D)." Although the correlation appears statistically significant (which is likely driven by the large sample size) the strength of the correlation is very weak (i.e. the biological significance is not very convincing).
12. Authors need to report more information on CD144 immunoprecipitation experiment including but not limited to how much serum was used per patient in this experiment to facilitate reproducibility.
13. Results Pg 8: The authors write, " Furthermore, the expression of miR-483-3p/-5p tend to be lower in lung ECs isolated from IPAH patients than HCs (Supplementary Fig. 3)." This statement is misleading by omission, as none of the data in the supplementary figures reached statistical significance.
14. Why were miR-483 levels in CD144-enriched EVs only assessed in a subset of the patients (i.e., 34-37 subjects versus 95-139 total subjects/group), and how were these patient/controls selected in an unbiased fashion?
15. The conclusions from the RNA-Seq experiment will be limited because the sample size of $n=2$ /group doesn't permit meaningful statistical analysis, so there is likely to be many false positive and negative errors associated with this experiment.
16. What does it mean to transfect the cells with miR-483-3p/-5p mimic? Was an equimolar mixture of both miRNAs used? This should be clarified in methods.
17. The authors suggest that decreased EC expression of miR-483 may be important for PAH. If so, why did the authors supplement (rather than inhibit) miR-483 levels in endothelial cells for the RNA-Seq experiment? The direct relevance of this experiment is questionable, particularly since the quantity of mimic delivered to the cell is probably not physiologically relevant (i.e., a gross excess of mimic).
18. Does Figure 2B show the most significant enriched GO terms (by p-value), or an arbitrary selection of significant GO terms among the 300 terms that were identified?
19. The sample sizes for each separate experiment in Figure 4 should be clearly reported. If $n=3$ per group, the experiments are probably statistically underpowered considering the variability and number of groups involved.
20. Pg 10. The authors write: "MCT administration induced these PAH-related genes in the wild-type but not in EC-miR-483-Tg rats". However, all of the protein levels of the genes in figure 4E appear to be increased in MCT-treated Tg rats versus MCT-treated WT rats, even though no stats are shown. This is hard to believe since there are smaller changes between the grey and white bars that reach statistical significance.
21. MiR-483-3p/-5p and related PAH genes should be measured in the serum and lung tissue of the Su/hypoxia rats to confirm their relevance in this animal model.

22. Figure 5D. Is there any quantitative summary data on the pulmonary vasculature, or is n=1/group in this angiography experiment?
23. Pg 11. The authors write, "The onset of PH, as revealed by mPAP, Fulton index, and angiography, was attenuated in EC-miR-483-Tg rats versus wild-type rats, both with SU5416/hypoxia treatment (Fig. 5F, G, H)." Again, this is an overstatement of the results as there was no significant difference in Fulton index between the EC-miR-483-Tg and WT rats treated with Su/hypoxia. In addition, the angiography data in Fig 5H only shows qualitative data for just one rat per group, which precludes robust conclusions. Of note, the Tg rats in Fig 5H appear to show greater vascular perfusion than the WT controls under baseline normoxia conditions, so this is a possible confounder for interpreting the results under Su/hypoxia conditions.
24. The Su/Hypoxia rat model typically leads to more severe changes than the MCT model. However, the hemodynamic and RV remodelling in Fig 5F & G (related to the Su/hypoxia model) appear at or below the levels observed in Fig 5A & B (related to MCT model). How do the authors reconcile this relatively mild Su/Hypoxia phenotype?
25. The authors need to report more detailed information on how the experiments in Figure 6 were conducted. For example, was the lentivirus delivered as a liquid bolus or aerosolized? What volume was used? What does orally mean precisely (i.e., was the lentivirus delivered via a catheter inserted orally into the trachea, or inhaled through the nares, or some other means)?
26. The authors need to clearly report if the sample size for ALL measurements in Figure 6 is n=7 per group. If not, the authors need to report the sample size for each applicable measurement. This confusion can be avoided with clearer figure legend descriptions or if the bar graphs are converted to dot plots.
27. Figure 6J. Again, n=1 rat/group for the angiography is not robust and the sample size should be increased to allow for quantification of the differences between groups.
28. How was wall thickness assessed in Figure 6I?
29. Pg 12. The authors write, "In line with the mitigated PH, Lenti-miR-483-GFP administration improved the survival of MCT-administered rats (Fig. 6K)." This is another example of a misleading overstatement, because no significant difference was observed between the groups. Furthermore, this non-significant difference between the groups is exaggerated by plotting the data on an abbreviated scale (i.e., from 70-100% instead of 0-100%).
30. What sample size was used in the survival experiment? Please confirm whether these are different or the same rats that were used in the other experiments in Figure 6? If the sample size is n=7/group, it's not clear how the survival curve was generated because the % survival don't seem consistent with this sample size. For example, if only 1 rat died you would at most have ~85% survival (if n=7), but the graph shows >90% survival for the M+483 group.
31. Did the authors conduct an experiment to test whether exogenously administered miR-483 will also attenuate Su/hypoxia-induced PH in rats? If this experiment was conducted and the data was negative, the authors should still report it in the manuscript.

Referee #2 (Comments on Novelty/Model System for Author):

Rat model of PAH is the best small animal model in the field. Novelty is high but some vascular analysis is not at a high quality (ex dye casting of vasculature).

Referee #2 (Remarks for Author):

The manuscript by Zhang et al. identifies that the amount of miR-483 is lower in the serum, and extracellular vesicles from patients with severe idiopathic pulmonary arterial hypertension (IPAH), suggesting that miR-483 can serve as a biomarker for IPAH. Administration of miR-483 mitigates the vascular remodeling and cardiopulmonary phenotypes of IPAH in rat PAH models (MCT and

Sugen/hypoxia) and in the transgenic rats in which miR-483 is expressed in the endothelium. Zhang et al. find that miR-483 targets multiple genes encoding the molecules in the TGF β signaling pathway, such as TGF- β , TGFBR2, suggesting that de-repression of this pathway contributes to the development of PAH phenotypes.

Generally speaking, the study includes lots of data, most experiments are well performed and the findings are novel and clinically significant. However, the study entirely relies on the overexpression of miR-483 and its therapeutic effect on PAH and lacks loss-of-function studies on miR-483. It is relatively easy to knock out a gene of interest by CRISPR-Cas9. It is important include miR-483 knock out animal study to prove a main point of the study. Furthermore, the study on miR-483 targets is limited to a validation of the seed sequence in miR-483 target mRNAs and fails to demonstrate a physiological significance of any miR-483 targets. Finally, some parts of the manuscript require further experiments and rewriting as stated below.

Major points:

1. Mice or rats genetically deleted in mR-483 or injected with antisense oligo. against miR-483 should be subjected to PAH models. This is absolutely critical to validate a main point of the study that a reduction of miR-483 is associated with severe PAH. What about the endothelial cells deleted or silenced in miR-483? Do they behave differently? What about the transcriptome of the endothelial cells +/- miR-483?

2. Cdh5 promoter drives embryonic endothelial expression, which precedes endothelial to hematopoietic transition. TGF- β signaling pathway, which is downregulated by miR-483 according to this study, is critical to both endothelial development and the development of hematopoietic progenitor and stem cells. It is necessary to investigate whether there is any developmental endothelial or hematopoietic defects in miR-483 transgenic (Tg) rats. For example, in Figure 5C, the saline treated miR-483 Tg rat show significantly dilated pulmonary artery compared to saline treated WT rat. Is there a vascular dilation in the miR-483 Tg rat? In Figure 5D, Saline treated miR-483 Tg rat exhibits increased angiogenesis compared with saline treated WT rat. Similarly, in Figure 5H, normoxia treated miR-483 Tg rat shows increased angiogenesis. Is this a phenotype?

3. The authors should perform experiments to interrogate how the pulmonary endothelium overexpressing miR-483 affect EC proliferation and apoptosis by BrdU staining and TUNEL staining of the Tg rats, respectively especially at early stages of PAH.

4. In Fig. 5D and Fig. 5H, the pulmonary artery is not casted, which is probably due to improper dissection of the lung. In Fig. 5D and Fig. 5H, WT lung after MCT or SU/Hypoxia treatment is not cleared properly, therefore, even the major vessels are not visible. The casting of lung vasculature should be repeated and all the tissues should be dissected and cleared properly.

5. miR-483 guide strand vs passenger strand are not discussed. The most direct way to tell guide strand from passenger strand is to show the abundance of these two strands. For example, in Fig. 3B, 3F and 4B, the relative fold of miR-483-3p and miR-483-5p to the internal control (U6?) or spike-in control or AGO amount should be presented in one plot, so the abundance of miR-483-3p and miR-483-5p will be clear.

6. Phospho-Smad2/3 immunoblot analysis should be performed to indicate the level of TGF β -Smad signaling pathway in miR-483 Tg vs WT.

7. The study should demonstrate a relevance of some of miR-483 targets to pulmonary vascular remodeling. For example, does overexpression of some of the miR-483 target ameliorate vascular remodeling? Improve EC function?

Minor point:

1. In the discussion, some data are not shown. Instead of "data not shown", authors can present all the supportive data in the supplementary figures.

2. The lung casting assay is not described properly in supplementary material & methods. There is only a sentence stating that "Microfil MV-122, MV-Diluent, and MV curing agent (Flow Tech) were used for pulmonary angiography as described by the manufacturer." The manufacturer describes how to mix, cure the Microfil and clear the tissues. How the authors perfuse microfil to the lung? Is it from inferior vena cava?

3. The wound healing assay (Figure 4D) is not described in the supplementary material&methods.

4. Acknowledgement is not properly shown (edited).

Referee #3 (Comments on Novelty/Model System for Author):

The authors used serum samples from health controls and patients with idiopathic pulmonary arterial hypertension (IPAH), as well as in vitro cell models and in vivo animal models in the study. I think it is adequate and appropriate, and a strength, that the authors used multiple models for the study.

Referee #3 (Remarks for Author):

Comments for the Authors

In the manuscript by Zhang et al., entitled "MicroRNA-483 ameliorates pulmonary arterial hypertension", the authors investigated the critical role of miR-483 the development and progression of pulmonary hypertension using combined in vitro, ex vivo and in vivo models. The authors found that the level of miR-483 was lower in serum from patients with idiopathic pulmonary arterial hypertension (IPAH) than in normal subjects. In in vitro experiments, overexpression of miR-483 in endothelial cells (ECs) inhibited inflammatory and fibrogenic responses, as evidenced by the decreased expression levels of TGF- β , TGFBR2, b-catenin, CTGF, IL-1 β , and ET-1. In in vivo experiments, the authors showed that EC-specific overexpression of miR-483 significantly reduced pulmonary arterial pressure and inhibited pulmonary vascular wall thickening and right ventricular hypertrophy in rats with MCT- and Hypoxia/Sugen-induced pulmonary hypertension. Based on these results, the authors concluded that reduced miR-483 plays an important pathogenic role in the development and progression of pulmonary hypertension and miR-483 could be used to treat inflammation-associated pulmonary hypertension.

The results from this study demonstrate a very interesting link between miR-483 and the development of experimental pulmonary hypertension in animal models. The experiments appeared to be well designed and carefully performed by a group of investigators who have extensive research experience and expertise in the field of vascular pathobiology. The data presented in the study are novel and significant, and of high quality. The evidence for the novel link is compelling given the combined use of samples from patients with IPAH and in vitro/in vivo models in the study. I have only a few minor concerns that may need authors' attention to improve the manuscript.

Minor comments:

1. PAH and PH seem to be used interchangeably in the manuscript. The results from the study does not actually show that "miR-483 ameliorates pulmonary arterial hypertension", as indicated by the title of the manuscript. The data from the study show that the reduced miR-483 level in serum is associated with PAH, while in vitro and in vivo experiments show that miR-483 is involved in inhibiting experimental pulmonary hypertension in animal models. Consider changing the title of the manuscript.
2. The correlation data using human serum are very interesting, however, the resource or the origin of the reduced miR-483 level in serum in IPAH patients are unclear. The authors may consider including more discussion on the potential mechanisms involved in the reduced miR-483 in serum and its relation to the downregulated level in pulmonary vascular endothelial cells.
3. Figure 1A: the data show results from 95 control subjects and 139 IPAH patients. Since the level of miR-483 overlaps in many control subjects and IPAH patients, it would be good to include a histogram graph to show the distribution of miR-483 serum level in control and IPAH.
4. For immunoblotting images (Figs. 3D, 4E, 6D), the molecular weight size in kDa should be indicated so the audience can evaluate the size of the target bands/proteins.
5. Are there any differences of the expression levels of miR-483-3p and miR-483-5p in lung ECs between healthy controls and IPAH patients (Figure 2) and in aortic ECs between WT rats and EC-miR-483-Tg rats?

Reviewer #1

1. *“When were the serum samples collected and over what time period? This information should be reported in the methods to provide some insight into the age-related quality of the samples. The information should be reported separately by IPAH patients and healthy controls to rule out the potential for underlying technical artifacts”*

In total, 139 IPAH patients were prospectively enrolled from June 2015 to February 2018 at FuWai Hospital, Chinese Academy of Medical Sciences. Patients diagnosed with IPAH met the published guidelines for the diagnosis of pulmonary hypertension (*Eur Heart J.* 2016; 37: 67-119), i.e. mPAP \geq 25 mmHg at rest, pulmonary artery wedge pressure \leq 15 mmHg, and pulmonary vascular resistance (PVR) $>$ 3 Wood units (*Lancet.* 2008; 371: 2093-2100). Patients were excluded if associated with a definite cause, including connective tissue disease, congenital heart disease, chronic pulmonary thromboembolism, and PAH due to left heart disease, lung diseases, and hypoxemia. A group of 95 age- and sex-matched healthy subjects were contemporaneously recruited from FuWai Hospital as well. The investigation conformed to the principles outlined in the Declaration of Helsinki and the Department of Health and Human Services Belmont Report. The study protocol was approved by the ethics committees of FuWai Hospital and written informed consent was obtained from all subjects. Blood samples were drawn from the cubital vein within one week from the time of right-heart catheterization. Blood were taken from IPAH patients or healthy subjects in the fasting state for approximately 12 hr. None of the individuals had exercise before blood collection. After centrifugation at 1500 \times g for 10 min, the serum was aliquoted into separator tubes, quickly frozen in liquid nitrogen, and stored at -80 °C until use (see revised manuscript, In 3-18, p. 13).

2. *“Please report the passage number(s) related to cell culture experiments”*

PAECs (Lonza, Basel, Switzerland) with passage 4-7 were used for all cell culture experiments.

3. *“At what level is miR-483-3p/-5p present in serum? Please report Cq values to provide some indication of the detection level in serum.”*

The Cq values of miR-483-3p were usually 29-32 and those of miR-483-5p were 30-34.

4. *“Please clarify how miR-483 levels in plasma were normalized? Was the data normalized to cel-miR-39 or U6. If normalized to U6, please report the detection level of U6 in the serum”*

We used cel-miR-39 to normalize miR-483 levels in serum, which is commonly used as internal control in measuring microRNAs in the circulation.

5. *“Please report the volume of serum per patient used for extraction”*

We used 200 μ l serum from each patient for RNA extraction, which is described in the Method section.

6. "Please report how much RNA per sample was used for reverse transcription and how was this standardized across samples?"

We used 500 ng RNA per sample for reverse PCR. An equal amount of total RNA was used to standardize among samples.

7. "What evidence can the authors provide to indicate that the quantity and quality of serum-extracted RNA was comparable between samples (especially between healthy and IPAH groups)? This is an important consideration since cel-miR-39 is a poor control (as compared to an internal reference control) since it only accounts for variations incurred after the spike-in event. How do the authors know the RNA in the IPAH serum samples was not degraded, leading to reduced miRNA levels?"

The quality of RNA was assessed by spectrophotometric A260/A280 ratio. Overall, there was no significant difference of A260/A280 ratios between serum samples from healthy and IPAH groups. We also detected the circulatory level of microRNAs which have not been reported to be affected by PAH. As shown below, neither miR-126 nor miR-92a was decreased in IPAH serum. This result also supports the notion that microRNAs might not be degraded in serum samples from IPAH patients.

8. "The difference in serum miR-483 levels between IPAH and healthy controls is very modest, suggesting it would be a poor biomarker. Moreover, there is little value in a biomarker that can distinguish between IPAH and healthy people (as opposed to more clinically relevant control groups.)"

We fully understand the lack of specific marker for PAH or pulmonary vascular remodeling. As stated in *Eur Heart J.* 2015;37:67-119, the available markers can be grouped into those of vascular dysfunction, inflammation, myocardial stress, markers CO and/or tissue hypoxia, and markers of secondary organ damage. We value very much your opinion, however, would like to respectively argue the purpose of this piece of data. Lower level of serum miR-483 might be due in part to the dysfunctional endothelium as shown by anti-CD 144 immunoprecipitation experiment (Fig. 2A). We report this result because of its mechanistic link to EC impairments, but not miR-483 as an IPAH biomarker.

9. "Results Pg 7: The author's statement that, "Patients with higher mPAP and PVR seemed to have lower miR-483-3p/-5p levels (Supplementary Fig. 1A, B)." is not supported by the data in supplementary Fig 1A and B. I don't see any consistent and clear differences between the stratified groups, so the data is at best inconclusive"

Thanks for your comments, we agreed with your critics and therefore have deleted supplementary Fig. 1A and 1B in the revised manuscript and the statement.

10. *"Results Pg 7: The author's statement that, "...the 3 determinants of prognosis according to 2015 ESC/ERC guidelines for PH [mean right atrial pressure (mRAP), Cardiac index (CI), WHO functional class, and N-terminal pro-brain natriuretic peptide (NT-proBNP)] were also related to miR-483-3p/-5p levels (Supplementary Fig. 1C-F)," is again not supported by the data in the supplementary figures. There doesn't appear to be any differential response related to clinical severity of PAH"*

Thanks for your comments, we agreed with your opinion and therefore have deleted supplementary Fig. 1C-1F and the statement in the revised manuscript.

11. *"Results Pg 7: Regarding the author's statement, "Interestingly, the serum levels of ET-1 and IL-6 were inversely correlated with those of miR-483-3p (Fig. 1D)." Although the correlation appears statistically significant (which is likely driven by the large sample size) the strength of the correlation is very weak (i.e. the biological significance is not very convincing"*

We agree with your comments and therefore have toned down the statement as "the serum levels of ET-1 and IL-6 seemed to be inversely correlated with those of miR-483-3p" (see revised manuscript, ln. 19-20, p. 5).

12. *"Authors need to report more information on CD144 immunoprecipitation experiment including but not limited to how much serum was used per patient in this experiment to facilitate reproducibility"*

We have included a more detailed description of the CD144 immunoprecipitation experiment as the following: "Briefly, CD144-enriched EVs were immunoprecipitated from human or rodent serum (1 mL) by IP with anti-CD144 antibody (Santa Cruz Biotechnology) (1 µg per sample) incubating with Dynabeads (Invitrogen). IgG was used as an isotype control of IP." (See revised manuscript, ln. 5-7, p. 14).

13. *"Results Pg 8: The authors write," Furthermore, the expression of miR-483-3p/-5p tend to be lower in lung ECs isolated from IPAH patients than HCs (Supplementary Fig. 3)." This statement is misleading by omission, as none of the data in the supplementary figures reached statistical significance"*

We agreed with your comment and therefore have deleted supplementary Fig. 3 accordingly.

14. *"Why were miR-483 levels in CD144-enriched EVs only assessed in a subset of the patients (i.e., 34-37 subjects versus 95-139 total subjects/group), and how were these patient/controls selected in an unbiased fashion?"*

This experiment included CD144-immunoprecipitation which required a minimum amount of 200 µl of serum sample. Only 34 of 37 healthy controls and 95 of 139 IPAH subjects had serum stock more than 200 µl. The demography of these individuals was shown in the revised Table 1.

15. *“The conclusions from the RNA-Seq experiment will be limited because the sample size of $n=2$ /group doesn't permit meaningful statistical analysis, so there is likely to be many false positive and negative errors associated with this experiment”*

In order to limit the false positive and negative errors, we have performed RNA-Seq experiments in HUVECs as independent repeats. As shown in the heat map in the following, genes involved in proliferation, migration, inflammation, TGF β receptor and Wnt signaling, apoptosis process, response to hypoxia, and oxidative stress were largely decreased by miR-483. This independent set of RNA-seq data was in agreement with experiments involving PAECs.

16. *“What does it mean to transfect the cells with miR-483-3p/-5p mimic? Was an equimolar mixture of both miRNAs used? This should be clarified in methods”*

We transfected cells with an equimolar of miR-483-3p and miR-483-5p. This information has been provided in Methods (see revised manuscript, ln. 11-12, p. 14).

17. *“The authors suggest that decreased EC expression of miR-483 may be important for PAH. If so, why did the authors supplement (rather than inhibit) miR-483 levels in endothelial cells for the RNA-Seq experiment? The direct relevance of this experiment is questionable, particularly since the quantity of mimic delivered to the cell is probably not physiologically relevant (i.e., a gross excess of mimic)”*

With the consideration of therapeutic implication, we opted to overexpress miR-483 at the supraphysiological level to investigate genes and pathways affected by miR-483.

18. *“Does Figure 2B show the most significant enriched GO terms (by p-value), or an arbitrary selection of significant GO terms among the 300 terms that were identified?”*

Because this study focuses on PH, so we arbitrarily selected significantly enriched GO terms which are related to PH.

19. *“The sample sizes for each separate experiment in Figure 4 should be clearly reported. If $n=3$ per group, the experiments are probably statistically underpowered considering the variability and number of groups involved”*

We pooled RNA or protein samples from 3 rats for three separate RT-PCR or Western analyses ($n=3 \times 3$). This information has been included in the legend of Fig. 4 (see revised manuscript, ln. 5-7, p. 32).

20. *“Pg 10. The authors write: “MCT administration induced these PAH-related genes in the wild-type but not in EC-miR-483-Tg rats”. However, all of the protein levels of the genes in figure 4E appear to be increased in MCT-treated Tg rats versus MCT-treated WT rats, even though no stats are shown. This is hard to believe”*

since there are smaller changes between the grey and white bars that reach statistical significance”

We rationalized that “MCT-treated Tg rats versus MCT-treated WT rats” in your comment would be “MCT-treated Tg rats versus saline-treated Tg rats”. To this end, results in Fig. 4E suggest that miR-483 ameliorated, but not abolished, PH pathogenesis. Therefore, we have toned down the sentence as “MCT administration induced these PAH-related genes to much higher levels in the wild-type than those in EC-miR-483-Tg rats (see revised manuscript, ln. 6-7, p. 8).

21. “MiR-483-3p/-5p and related PAH genes should be measured in the serum and lung tissue of the Su/hypoxia rats to confirm their relevance in this animal model”

In accordance with the reviewer’s comment, we have measured levels of miR-483-3p/5p and PAH-related genes (e.g., TGF β , TGFBR2, CTGF, β -catenin, IL-1 β , and ET-1) in the serum and lung tissue of the Su/hypoxia rats (data are shown below). This set of new data have been included in revised manuscript, ln. 6-9, p. 9, and revised appendix Fig. S5.

22. “Figure 5D. Is there any quantitative summary data on the pulmonary vasculature, or is n=1/group in this angiography experiment?”

Because of the technical difficulties to optimize angiographic images of rat lung, we opted to remove the angiography data from Fig. 5. This omission does not change the conclusion of this figure.

23. “Pg 11. The authors write, “The onset of PH, as revealed by mPAP, Fulton index, and angiography, was attenuated in EC-miR-483-Tg rats versus wild-type rats, both with SU5416/hypoxia treatment (Fig. 5F, G, H).” Again, this is an overstatement of the results as there was no significant difference in Fulton index between the EC-miR-483-Tg and WT rats treated with Su/hypoxia. In addition, the angiography data in Fig 5H only shows qualitative data for just

one rat per group, which precludes robust conclusions. Of note, the Tg rats in Fig 5H appear to show greater vascular perfusion than the WT controls under baseline normoxia conditions, so this is a possible confounder for interpreting the results under Su/hypoxia conditions"

The P values for mPAP and Fulton index between the EC-miR-483-Tg and WT rats treated with Su/hypoxia were 0.02 and 0.06, respectively. We have modified the description as "The onset of PH, as revealed by increased mPAP and Fulton index tended to be attenuated in EC-miR-483-Tg versus wild-type rats, both with SU5416/hypoxia treatment" (see the revised manuscript, ln. 4-6, p. 9). The angiography images were also removed.

- 24.** *"The Su/Hypoxia rat model typically leads to more severe changes than the MCT model. However, the hemodynamic and RV remodelling in Fig 5F & G (related to the Su/hypoxia model) appear at or below the levels observed in Fig 5A & B (related to MCT model). How do the authors reconcile this relatively mild Su/Hypoxia phenotype?"*

Whether Su/Hypoxia treatment leads to more severe PH than MCT remains elusive. Even under the standard protocol (also used in this study), severity of PH caused by Sugen+hypoxia varied among laboratories. For example, Savai *et al.* reported a lower RVSP in Sugen/Hypoxia-treated PH rats than that in MCT-treated counterparts (*Nat Med.* 2014; 20: 1289-1300). As our study focuses on the comparison of PH between WT and Tg rats, not the severity between different PH models, our data did show that Tg+Su/hypoxia rats had ameliorated mPAP compared to WT+Su/hypoxia ones.

- 25.** *"The authors need to report more detailed information on how the experiments in Figure 6 were conducted. For example, was the lentivirus delivered as a liquid bolus or aerosolized? What volume was used? What does orally mean precisely (i.e., was the lentivirus delivered via a catheter inserted orally into the trachea, or inhaled through the nares, or some other means)?"*

We have added detailed information in the Methods section as the following "Briefly, rats were anesthetized. Then, curved blunted-ended forceps were used to grasp the tongue to gain visualization of the larynx. The lentivirus (1×10^9 PFU), diluted in 100 μ L sterile PBS, were instilled using a PE-10 tubing inserted into the trachea" (see revised manuscript, ln. 19-22, p. 15).

- 26.** *"The authors need to clearly report if the sample size for ALL measurements in Figure 6 is n=7 per group. If not, the authors need to report the sample size for each applicable measurement. This confusion can be avoided with clearer figure legend descriptions or if the bar graphs are converted to dot plots"*

In accordance with the reviewer's suggestion, the sample size for each measurement was described in the legend of Fig. 6 (see revised manuscript, ln. 1-2 and ln. 6, p. 33).

- 27.** *"Figure 6J. Again, n=1 rat/group for the angiography is not robust and the sample size should be increased to allow for quantification of the differences between groups"*

Please refer to Q22. We have omitted the angiographic data from Fig. 6.

- 28.** *"How was wall thickness assessed in Figure 6I?"*

The wall thickness was assessed according the commonly used methods (*Nat Med.* 2014; 20: 1289-1300). The following information has been added to the Methods (see revised manuscript, ln. 5-8, p. 16) “we used the picture scaleplate to measure the thickness of pulmonary arteries (20-70 μm in diameters, $n=10$ for each rat). The external diameter (D_{ex}) and inner diameter (D_{in}) of each artery were measured, and then the wall thickness was assessed using $(D_{\text{ex}} - D_{\text{in}})/D_{\text{ex}} * 100\%$ ”.

29. “Pg 12. The authors write, “In line with the mitigated PH, Lenti-miR-483-GFP administration improved the survival of MCT-administered rats (Fig. 6K).” This is another example of a misleading overstatement, because no significant difference was observed between the groups. Furthermore, this non-significant difference between the groups is exaggerated by plotting the data on an abbreviated scale (i.e., from 70-100% instead of 0-100%)”

The difference of survival between MCT+lenti-miR-483-GFP and MCT+lenti-GFP rats was with a P value of 0.058. We have revised the sentence as “In line with the mitigated PH, Lenti-miR-483-GFP administration seemed to improve the survival of MCT-administered rats” (see revised manuscript, ln. 21-22, p. 9).

In accordance with the reviewer’s suggestion, we have changed the abbreviated scale to 0-100% and labeled the P value in the revised Fig. 6J.

30. “What sample size was used in the survival experiment? Please confirm whether these are different or the same rats that were used in the other experiments in Figure 6? If the sample size is $n=7$ /group, it's not clear how the survival curve was generated because the % survival don't seem consistent with this sample size. For example, if only 1 rat died you would at most have ~85% survival (if $n=7$), but the graph shows >90% survival for the M+483 group”

We used 15 rats in each group. Different groups of rats were used for the survival experiment. This information has been added to the legend of Fig. 6 (see revised manuscript, ln. 6, p. 33).

31. “Did the authors conduct an experiment to test whether exogenously administered miR-483 will also attenuate Su/hypoxia-induced PH in rats? If this experiment was conducted and the data was negative, the authors should still report it in the manuscript”

Given the beneficial effect of miR-483 in PH had been shown by the use of EC-miR-483-Tg rats receiving Su/hypoxia treatment, we opted not to deliver lenti-miR-483 to Su/hypoxia-treated rats.

Reviewer #2

Major point:

1. "Mice or rats genetically deleted in miR-483 or injected with antisense oligo. against miR-483 should be subjected to PAH models. This is absolutely critical to validate a main point of the study that a reduction of miR-483 is associated with severe PAH. What about the endothelial cells deleted or silenced in miR-483? Do they behave differently? What about the transcriptome of the endothelial cells +/- miR-483?"

In response to your valuable comments, we have used miR-483 antagomir to inhibit miR-483 in rat lung for examining the effect of decreased miR-483 on PH pathogenesis. As shown in the figure below, MCT-treated rats receiving miR-483 antagomir showed increased mPAH and Fulton index, when compared with MCT-PH rats receiving scramble miR. These new results suggest that miR-483 inhibition exacerbates PH pathogenesis and hence it has been included in the revised manuscript, ln. 19-24, p. 8-ln. 1-2, p. 9, and appendix Fig. S4.

We also inhibited miR-483 in PAECs, as shown below, anti-miR-483 led to the increase in the expression of miR-483 target genes. These data have been included in the revised manuscript, ln. 1-3, p. 7, and appendix Fig. S2.

Because of a 90-day period of time was allowed for revision, we did not perform new RNA-seq experiment on endothelial cells +/- miR-483. However, existing RNA-seq data on HUVECs transfected with miR-483 reveal that genes involved in proliferation, migration, inflammation, TGF β receptor and Wnt signaling, apoptosis process, response to hypoxia, and oxidative stress were grossly decreased by miR-483. This independent set of RNA-seq data was in agreement with experiments involving PAECs.

2. “*Cdh5* promoter drives embryonic endothelial expression, which precedes endothelial to hematopoietic transition. TGF- β signaling pathway, which is downregulated by miR-483 according to this study, is critical to both endothelial development and the development of hematopoietic progenitor and stem cells. It is necessary to investigate whether there is any developmental endothelial or hematopoietic defects in miR-483 transgenic (Tg) rats. For example, in Figure 5C, the saline treated miR-483 Tg rat show significantly dilated pulmonary artery compared to saline treated WT rat. Is there a vascular dilation in the miR-483 Tg rat? In Figure 5D, Saline treated miR-483 Tg rat exhibits increased angiogenesis compared with saline treated WT rat. Similarly, in Figure 5H, normoxia treated miR-483 Tg rat shows increased angiogenesis. Is this a phenotype?”

As shown in appendix Fig. S3, of the level of miR-483 was not drastically elevated in the peripheral blood mononuclear cells isolated from EC-miR-483-Tg rats, when compared with their wild-type littermates. This result indicates the tissue-specific expression of miR-483 in our EC-miR-483-Tg rats.

To further explore the potential impact of miR-483 in developmental defects of hematopoietic cells, we have evaluated the cell counts of red blood cell, white blood cell, and blood platelet in rats circulating, as shown in table below,

Hematopoietic character of WT and Tg rats

	WT (n=3)	Tg (n=3)
Red Blood Cell (10^{12} /L)	7.71±1.13	7.90±0.63
White Blood Cell (10^{12} /L)	778.3±276.4	565.3±276.4
Blood Platelet (10^9 /L)	9.5±5.0	9.9±1.3
Lymphocyte (10^9 /L)	7.0±4.1	7.0±2.1
Monocyte (10^9 /L)	0.97±0.49	1.03±0.57
Neutrophil (10^9 /L)	1.46±0.73	1.79±0.37
EO (10^9 /L)	0.05±0.02	0.09±0.07
BASO (10^9 /L)	0.03±0.03	0.03±0.03

To verify if any effect of miR-483 on vessel dilation, vasorelaxation experiment was performed using Multi Myograph system. As shown in figure below, a comparable vasodilation was found between aortae isolated from EC-miR-483-Tg rats and their WT littermates.

3. *“The authors should perform experiments to interrogate how the pulmonary endothelium overexpressing miR-483 affect EC proliferation and apoptosis by BrdU staining and TUNEL staining of the Tg rats, respectively especially at early stages of PAH”*

We have isolated lung ECs from early stages (10 days after receiving MCT injection) of PH rats. BrdU staining and TUNEL staining were performed to assess the proliferation and apoptosis of ECs, respectively. Results shown below indicate that miR-483 overexpression suppressed EC proliferation and apoptosis at early stage of PH in rat models.

4. *“In Fig. 5D and Fig. 5H, the pulmonary artery is not casted, which is probably due to improper dissection of the lung. In Fig. 5D and Fig. 5H, WT lung after MCT or SU/Hypoxia treatment is not cleared properly, therefore, even the major vessels are not visible. The casting of lung vasculature should be repeated and all the tissues should be dissected and cleared properly”*

Because of the technical difficulties to optimize angiographic images of rat lung, we opted to remove the angiography data from Fig. 5. This omission does not change the conclusion of this figure.

5. “miR-483 guide strand vs passenger strand are not discussed. The most direct way to tell guide strand from passenger strand is to show the abundance of these two strands. For example, in Fig. 3B, 3F and 4B, the relative fold of miR-483-3p and miR-483-5p to the internal control (U6?) or spike-in control or AGO amount should be presented in one plot, so the abundance of miR-483-3p and miR-483-5p will be clear”

According to your suggestions, we have shown the relative fold of miR-483-3p and -5p in Fig. 3B, 3F and 4B.

6. “Phospho-Smad2/3 immunoblot analysis should be performed to indicate the level of TGFb-Smad signaling pathway in miR-483 Tg vs WT”

We have performed Western blotting to detect p-Smad2 levels using rat lung samples and found that Tg rats showed decreased phospho-Smad2 protein level compared to WT ones receiving MCT.

7. “The study should demonstrate a relevance of some of miR-483 targets to pulmonary vascular remodeling. For example, does overexpression of some of the miR-483 target ameliorate vascular remodeling? Improve EC function?”

Reviewer 2 probably meant overexpression of miR-483 targets “potentiate” rather than “ameliorate” vascular remodeling. Most, if not all, genes involved in our study have been reported to be involved in vascular remodeling in the context of PAH (*Nat Rev Cardiol.* 2011; 8: 443-455). For example, Calvier *et al.* reported that TGF- β induces PASMC proliferation and fibrosis and promotes

vascular remodeling during PAH pathogenesis (*Cell Metab.* 2017; 25: 1118-1134); Takahashi *et al.* found that β -catenin is activated and associated with abnormal PSMC proliferation (*Fed Eur Biochem Soc.* 2016; 590: 101-109); Humbert *et al.* showed elevated levels of IL-1 β and ET-1 in the circulation of PAH patients (*Am J Respir Crit Care Med.* 1995; 151: 1628-1631; Giaid *et al.* *N Engl J Med.* 1993; 328: 1732-1739). Satwiko *et al.* demonstrated that the ET-1 transgenic mice developed severer PH (*Biochem Biophys Res Commun.* 2015; 465: 356-362).

Minor point:

1. "In the discussion, some data are not shown. Instead of "data not shown", authors can present all the supportive data in the supplementary figures"

We have included these data in the revised appendix Fig. S13 (see revised manuscript, In. 23, p. 11).

2. "The lung casting assay is not described properly in supplementary material & methods. There is only a sentence stating that "Microfil MV-122, MV-Diluent, and MV curing agent (Flow Tech) were used for pulmonary angiography as described by the manufacturer." The manufacturer describes how to mix, cure the Microfil and clear the tissues. How the authors perfuse microfil to the lung? Is it from inferior vena cava?"

For the procedure, we opened the thoracic cavity of rats, ligated the inferior vena cava, aorta and superior vena cava, and injected the mixed microfil into right ventricular. Because the quality of angiography was not optimal, we have deleted the angiography data.

3. "The wound healing assay (Figure 4D) is not described in the supplementary material&methods"

The detailed method of wound healing assay has been added to the revised methods (see revised manuscript, In. 9-14, p. 17).

4. "Acknowledgement is not properly shown (edited)"

We have revised the section of Acknowledgement.

Referee #3

Minor comments:

1. *“PAH and PH seem to be used interchangeably in the manuscript. The results from the study does not actually show that “miR-483 ameliorates pulmonary arterial hypertension”, as indicated by the title of the manuscript. The data from the study show that the reduced miR-483 level in serum is associated with PAH, while in vitro and in vivo experiments show that miR-483 is involved in inhibiting experimental pulmonary hypertension in animal models. Consider changing the title of the manuscript”*

We have changed the title of the manuscript as “MicroRNA-483 Amelioration of Experimental Pulmonary Hypertension”.

2. *“The correlation data using human serum are very interesting, however, the resource or the origin of the reduced miR-483 level in serum in IPAH patients are unclear. The authors may consider including more discussion on the potential mechanisms involved in the reduced miR-483 in serum and its relation to the downregulated level in pulmonary vascular endothelial cells”*

We reasoned that the decreased miR-483 levels were due at least in part to the decreased expression or activity of the upstream molecules, such as KLF2 and AMPK. Circulating exosome has been reported to be increased in PH patients and animals, so the decreased miR-483 in serum might be resulted from decreased miR-483 levels in ECs. While we appreciate your points, detailed mechanisms warrant further investigation.

3. *“Figure 1A: the data show results from 95 control subjects and 139 IPAH patients. Since the level of miR-483 overlaps in many control subjects and IPAH patients, it would be good to include a histogram graph to show the distribution of miR-483 serum level in control and IPAH”*

We have added a histogram graph to show the miR-483 levels in control and IPAH patients (see revised manuscript, Fig. 1A, Fig. 1C, and Fig. 2A).

4. *“For immunoblotting images (Figs. 3D, 4E, 6D), the molecular weight size in kDa should be indicated so the audience can evaluate the size of the target bands/proteins”*

We have added the molecular weight size in all immunoblotting images.

5. *“Are there any differences of the expression levels of miR-483-3p and miR-483-5p in lung ECs between healthy controls and IPAH patients (Figure 2) and in aortic ECs between WT rats and EC-miR-483-Tg rats?”*

We have tested the expression levels of miR-483-3p and -5p in lung ECs isolated from IPAH patients and found they tended to be decreased compared to healthy controls. Also, EC-miR-483-Tg rats had elevated miR-483 levels in aortic ECs compared to WT controls (Appendix Fig. S3).

2nd Editorial Decision

14th January 2020

Thank you for the submission of your revised manuscript to EMBO Molecular Medicine. We have now received the enclosed report from the three reviewers who were asked to re-assess it. As you will see while the reviewer #2 and #3 are now overall supportive, reviewer #1 still raises a couple of concerns on your work.

In principle, our editorial policy only allows a single round of major revision. However, as EMBO Molecular Medicine values reliable and reproducible research, which requires transparent and detailed reporting of data and methods, we think it is important to address reviewer #1's concerns with regard to the RNA quality assessment and data analysis. We would therefore ask you to address these points together with other comments from reviewer #1 in an exceptional second round of revision. The revised manuscript will be reassessed by reviewer #1, and acceptance or rejection of the manuscript will depend on another round of review. Therefore, your responses should be as complete as possible.

***** Reviewer's comments *****

Referee #1 (Comments on Novelty/Model System for Author):

See comments for the reviewers.

The manuscript has been revised for the second time; however, there are still several substantive issues impacting the interpretation of data and reliability of some conclusions. Below I have detailed the main deficiencies in the response to my previous concerns.

I still believe the remaining concerns can be addressed, assuming that they are in fact using an appropriate statistical approach in dealing with technical replicates; however, it is disappointing that they were not more transparent in addressing these concerns in the first place and I am not sure it is warranted to give them a third try.

Referee #1 (Remarks for Author):

The manuscript has been revised for the second time; however, there are still several substantive issues impacting the interpretation of data and reliability of some conclusions. Below I have detailed the main deficiencies in the response to my previous concerns.

Previous comment 7: I do not believe the authors have provided sufficient evidence to demonstrate that the quality of the serum-extracted RNA was comparable between the IPAH and control groups.

The spectrophotometric assessment of A260/A280 is inadequate because it only provides insight into the purity of the RNA (specifically whether the RNA is contaminated with proteins), and no data is actually shown. Furthermore, the authors did not address the A260/A230 ratio, which is another key ratio used to assess RNA purity (e.g., for lysis/wash buffer contamination). Importantly, these ratios provide no information on RNA integrity, which is a separate concept from purity. For example, the RNA could be 100% pure RNA (as reflected by 'good' ratios), but still be severely degraded into smaller RNA fragments, leading to lower miRNA levels. The measurement of miR-126 and miR-92a as potential internal controls is a good idea, but the execution is incomplete because only 9-10 subjects (from a total of 95-139 patients) were measured in each group. Therefore, there is a strong chance for selection bias, so the conclusion related to the observed serum differences in miR-483 in PAH patients is not well supported.

Previous comment 15: The authors response to this concern is not adequate. The addition of an RNA-Seq experiment using HUVECs with $n=1/\text{group}$ is only of marginal value and fails to address the issue of small sample size raised in the original comment. It's not clear how many total transcripts were assessed/detected in their RNA-seq experiment; however, one would expect to detect differences in 5% of those transcripts just by chance. For example, if 20,000 transcripts were detected, one would expect to find ~1,000 differentially altered transcripts just by chance, so the design of this experiment may yield unreliable results. Another relevant issue is that only an arbitrary selection of PH-related genes is shown, with no information on the number and type of other non-PH related transcripts that were 'altered' in this experiment. Therefore, I remain concerned that these results are presented in a possibly biased manner.

Previous comment 18: Again, I am unconvinced by the authors response to this comment. In their response, the authors admit to arbitrarily selecting the significantly enriched GO terms that are only related to PH. This begs the question as to whether, and how many, other non-PH related GO terms are significantly enriched (and not reported). This is problematic because it is biased and serves to frame the data so as to support their hypothesis. Furthermore, the legend of Figure 2B simply reports that "the top 300 up-regulated or down-regulated genes" are shown with no mention of the arbitrary selection procedure, which is misleading.

Previous comment 19: The authors response to this comment was unclear and potentially problematic. The authors report that "we pooled RNA or protein samples from 3 rats for three separate RT-PCR or Western analyses ($n=3 \times 3$)". This statement and sample size nomenclature is confusing because i) why are they pooling the rat samples prior to analysis and ii) how are they performing statistical analyses on pooled samples? As currently written, the author's response suggests that RNA or protein was pooled from 3 rats/group into 1 biological replicate/group, which was then measured 3 times. Another possible interpretation is that the authors performed statistical analysis on all biological AND technical replicates so $n=3 \times 3 = 9$ data points per group. Either way, the implication is that the authors may have biased the statistical analyses with technical replicates, which make the results less reliable. The final statistical analyses should only be performed with the biological replicates (i.e., $n=3$ rats/group, where the value for each biological replicate is the mean of the technical replicates).

Previous comment 26: The author's response to this comment is also unclear for the same reason as comment #19. It is a major problem if the authors are including technical replicates directly in their statistical analyses, rather than proper biological replicates.

Referee #2 (Remarks for Author):

The authors addressed all my concerns in the revised manuscript.

Referee #3 (Comments on Novelty/Model System for Author):

The models used in the study are all fine.

Referee #3 (Remarks for Author):

The authors have adequately and appropriately addressed my concerns and questions with new data and revision in the manuscript. I think the revised manuscript is deemed ready for publication in the journal.

“Previous comment 7: I do not believe the authors have provided sufficient evidence to demonstrate that the quality of the serum-extracted RNA was comparable between the IPAH and control groups. The spectrophotometric assessment of A260/A280 is inadequate because it only provides insight into the purity of the RNA (specifically whether the RNA is contaminated with proteins), and no data is actually shown. Furthermore, the authors did not address the A260/A230 ratio, which is another key ratio used to assess RNA purity (e.g., for lysis/wash buffer contamination). Importantly, these ratios provide no information on RNA integrity, which is a separate concept from purity. For example, the RNA could be 100% pure RNA (as reflected by 'good' ratios), but still be severely degraded into smaller RNA fragments, leading to lower miRNA levels. The measurement of miR-126 and miR-92a as potential internal controls is a good idea, but the execution is incomplete because only 9-10 subjects (from a total of 95-139 patients) were measured in each group. Therefore, there is a strong chance for selection bias, so the conclusion related to the observed serum differences in miR-483 in PAH patients is not well supported”

We agree that the miRNA integrity is essential to draw a reliable conclusion. In accordance with the Reviewer's suggestion, we have performed further experiments with increased number of measurements of miR-126 and miR-92a as potential internal controls. This new experiment contained almost all sera from IPAH patients and healthy control (HC) included in the original assays (93/139 IPAH; 95/95 HC). The few omitted was due to their low volumes not enough for miRNA isolation. As shown in the panel (A) below, comparable levels of miR-126 and miR-92a were found between IPAH and HC groups. Moreover, panel (B) indicates neither miR-126 nor miR-92a showed any correlation with miR-483. These results suggest that the differential levels of miR-483 in IPAH versus HC were not due to a different miRNA degradation rate between the two groups. Additional evidence supporting this argument was the lower serum levels of miR-483 also found in PH rodent models. In those animal experiments, sera were freshly collected (see revised Fig. 6B).

Although miRNAs are known to be remarkably stable in serum or plasma after a long-term storage (*Proc Natl Acad Sci U S A*, 2008;105:10513-10518, *PLoS ONE*, 2008;3:e3148), we strictly followed the protocols to ensure the microRNA integrity and avoid the selection bias. These standard procedures are commonly used by investigators in the field of cardiovascular sciences (*Circ Res*, 2010;107:677-684, *Circulation*, 2011;124:175-184). Moreover, serum samples from IPAH patients and healthy controls in our study were isolated, stored, and processed under the same conditions.

“Previous comment 15: The authors response to this concern is not adequate. The addition of an RNA-Seq experiment using HUVECs with $n=1/\text{group}$ is only of marginal value and fails to address the issue of small sample size raised in the original comment. It's not clear how many total transcripts were assessed/detected in their RNA-seq experiment; however, one would expect to detect differences in 5% of those transcripts just by chance. For example, if 20,000 transcripts were detected, one would expect to find ~1,000 differentially altered transcripts just by chance, so the design of this experiment may yield unreliable results. Another relevant issue is that only an arbitrary selection of PH-related genes is shown, with no information on the number and type of other non-PH related transcripts that were 'altered' in this experiment. Therefore, I remain concerned that these results are presented in a possibly biased manner”

The HUVEC RNA-seq experiments were performed 4 years ago. Providing the informative and comprehensive usefulness of big data analyses, this approach involving RNA-seq was intended for hypothesis generation, and then validated by wet experiments. Deduced from this piece of RNA-seq data, these miR-483-regulated genes reported in this manuscript were verified in the performed in vitro and in vivo experiments. The anti-fibrogenic and anti-inflammatory nature of miR-483 are further confirmed by results from other studies (*Circ Res*, 2017;120:354-365, *Arterioscler Thromb Vasc Biol*, 2019;39:467-481, *J. Cell. Mol. Med.*, 2014;18:966-974).

To date, the two biologic replicates in RNA-seq experiments are still commonly used (*J Clin Invest*, 2017;127:2555-2568, *J Clin Invest*, 2019;129:4492-4505). In our obtained data, 93408204, 96199076, 89861720 and 99060724 transcripts were detected in 2 controls and 2 miR-483 mimic samples, respectively. To further address the concern “these results are presented in a possibly biased manner”, we have reanalyzed the results by comparing our PAEC RNA-seq data with those PAEC data matrix available online (GEO: GSE78540). These data from independent source were used as untreated controls and the combined [2 controls (GEO: GSM2072328, GSM2072329) and 2 miR-483 mimic samples (from our own data)] heatmap was shown below. Through this “non-biased” analysis using data from independent source, miR-483 overexpressing remained caused a lower expression of genes involved in PAH pathogenesis (e.g., CTNNB1, EDN1, and SMAD2) in hPAECs. These results support our hypothesis from the original analysis that the miR-483 levels are related to the pathogenesis of PAH.

“Previous comment 18: Again, I am unconvinced by the authors response to this comment. In their response, the authors admit to arbitrarily selecting the significantly enriched GO terms that are only related to PH. This begs the question as to whether, and how many, other non-PH related GO terms are significantly enriched (and not reported). This is problematic because it is biased and serves to frame the data so as to support their hypothesis. Furthermore, the legend of Figure 2B simply reports that “the top 300 up-regulated or down-regulated genes” are shown with no mention of the arbitrary selection procedure, which is misleading”

In response to the reviewer’s comment, we have included full GO terms (e.g., cell cycle, signal transduction, protein stabilization, and autophagy pathway, etc., as shown in below) in Figure S2. Indeed, miR-483 overexpression in cultured ECs altered gene expressions linked to multiple functional outcomes. As mentioned above, based on the research goal of this study, we selected the biological pathways that are related to PH. Noticeably, miR-483 targeting of the TGF-β pathway was previously reported in several independent studies (*Tohoku J. Exp. Med.*, 2017;243:41-48; *J. Cell. Mol. Med.*, 2014;18:966-974), which supports the central hypothesis of our work.

The detailed procedure for data analyses is now included in the legend of Figure 2.

“Previous comment 19: The authors response to this comment was unclear and potentially problematic. The authors report that “we pooled RNA or protein samples from 3 rats for three separate RT-PCR or Western analyses (n=3x3)”. This statement and sample size nomenclature is confusing because i) why are they pooling the rat samples prior to analysis and ii) how are they performing statistical analyses on pooled samples? As currently written, the author’s response suggests that RNA or protein was pooled from 3 rats/group into 1 biological replicate/group, which was then measured 3 times. Another possible interpretation is that the authors performed statistical analysis on all biological AND technical replicates so n=3 x 3 = 9 data points per group. Either way, the implication is that the authors may have biased the statistical analyses with technical replicates, which make the results less reliable. The final statistical analyses should only be performed with the biological replicates (i.e., n=3 rats/group, where the value for each biological replicate is the mean of the technical replicates)”

We would like to correct the misled information in the last version of the manuscript. In fact, nine rats were used in each group in experiments reported in Figure 4. We randomly pooled three rats into one biological sample. A total number of 3 biological samples were used for qPCR, Western blot, and Ago-IP experiments. The statistical analyses were performed based on these three biological samples, not the three technical replicates.

Because the harvested rat lungs were used for angiography, Evans blue staining, the Ago-1 and Ago-2 IPs, RT-PCR, and western blot analyses, we economized the amounts of tissues for multiple assays by pooling samples from 3 animals.

“Previous comment 26: The author’s response to this comment is also unclear for the same reason as comment #19. It is a major problem if the authors are including technical replicates directly in their statistical analyses, rather than proper biological replicates”

Identical to the response above, we did not use technical replicates for statistical analysis. Three independent groups contained 9 different rats were used for the statistical analysis.

3rd Editorial Decision

9th March 2020

Thank you for the submission of your revised manuscript to EMBO Molecular Medicine. We have now received the enclosed report from referee #1 who raised a couple of issues in the last review. As you will see referee #1 is now supportive and I am pleased to inform you that we will be able to accept your manuscript pending the following amendments:

***** Reviewer's comments *****

Referee #1 (Remarks for Author):

The authors have adequately addressed the concerns raised in the last review.

3rd Revision - authors' response

13th March 2020

The Authors have made the requested editorial changes.

Accepted

17th March 2020

Please find enclosed the final reports on your manuscript. We are pleased to inform you that your manuscript is accepted for publication and is now being sent to our publisher to be included in the next available issue of EMBO Molecular Medicine.

YOU MUST COMPLETE ALL CELLS WITH A PINK BACKGROUND ↓
PLEASE NOTE THAT THIS CHECKLIST WILL BE PUBLISHED ALONGSIDE YOUR PAPER

Corresponding Author Name: Zhi-cheng Jing, John Y-J Shyy

Manuscript Number: EMM-2019-11303-V2